# Chemical controls on iron distributions across the subsurface South Pacific Ocean

M. Gledhill [1] ✉, K. Gosnell [1], M. P. Humphreys [2], L. Delaigue [2,3], N. Helle[1], K. Zhu[1], P. Lodeiro [4], C. Rey-Castro [4] & E. P. Achterberg [1]

Iron and nitrogen are the primary nutrients that limit productivity in the ocean. While nitrogen cycling is largely controlled by biology, iron cycling is strongly determined by chemistry because iron losses are driven by abiotic formation of authigenic mineral iron. Here, we apply a mechanistic approach to examine how organic matter across the dissolved-particulate size spectrum controls authigenic iron formation in subsurface waters (>250 m) of the South Pacific Ocean. We find that accounting for the chemical heterogeneity of organic matter is essential for predicting widespread authigenic iron formation. Predicted dissolved and particulate iron concentrations matched observations in the ocean interior, while discrepancies were linked to kinetic control of authigenic iron formation or inputs of particles from the seafloor. Our work shows the need to represent the complexity of abiotic interactions to better resolve the interplay of chemical and biological controls on ocean iron cycling.

Iron (Fe) is the fourth most abundant element on Earth, yet seawater concentrations are usually below 10 nmol L⁻¹. The low concentrations result from a low Fe solubility in the slightly basic (pH > 7), oxygenated ocean waters, and may lead to Fe limitation of phytoplankton growth[1,2]. Model simulations have shown that chemical (i.e., abiotic) drivers of Fe cycling are critical to an accurate description of Fe biogeochemistry in the upper ocean[3]. However, the processes that maintain such low concentrations are often described in simplified, operationally defined, black-box terms such as ligand binding[4] and scavenging[5,6].

In oxygenated seawater Fe binds to hydroxide ions, which results in authigenic Fe precipitation as amorphous ferric oxy-hydroxides (authFeOH)[7,8]. Organic matter competes with hydroxide ions for Fe binding and thus raises the solubility of Fe, hence it is important to accurately constrain this competitive interaction[9]. While binding of Fe to hydroxide ions and formation of authFeOH can be described with well-defined thermodynamic expressions[7], Fe binding to organic ligands is usually represented via simple approximations[4]. These approximations summarise Fe binding sites to one or two classes, typically termed strong and weak ligands, each with an empirically defined binding affinity, even though dissolved organic matter (DOM)

is a heterogeneous pool of binding sites with a wide range of affinities[2,10–13]. Furthermore, this representation of ligands cannot account for changes in ambient physicochemical conditions such as pH or temperature[14,15] that are expected to change in the future ocean[16].

The main abiotic pathway for Fe removal from the ocean is its incorporation into particles that sink out of the water column, in a process broadly described as scavenging[3,17]. Particulate phases comprise multiple forms[18,19], including authFeOH, particulate organic matter (POM), lithogenic matter, biogenic carbonates and silicates, and other authigenic metal (hydr)oxides. The most critical phases for Fe are the refractory lithogenic phase, authFeOH, and Fe bound to POM[20–23]. Precipitation of authFeOH and abiotic scavenging onto different particulate phases can be empirically related to the concentrations of soluble inorganic Fe (Fe′)[3,24,25], with Fe′ controlled by the competitive interactions between the hydroxide ion and organic matter (Fig. 1a)[3,26]. In biogeochemical models, ligand binding is a strong competitor for Fe relative to the hydroxide ion[27] and thus limits authFeOH precipitation, which can lead to overestimation of DFe[3,27]. Hence, Tagliabue et al.[3] introduced an empirically defined colloidal shunt to aggregate a portion of the DFe pool to the particulate phase.

[1]GEOMAR Helmholtz Centre for Ocean Research Kiel, Kiel, Germany. [2]NIOZ Royal Netherlands Institute for Sea Research, Department of Ocean Systems, Texel, the Netherlands. [3]Sorbonne Université, CNRS, Laboratoire d'Océanographie de Villefranche, Villefranche-Sur-Mer, France. [4]Department of Chemistry, Physics and Environmental and Soil Sciences, University of Lleida and AGROTECNIO-CERCA Centre, Lleida, Spain. ✉e-mail: mgledhill@geomar.de

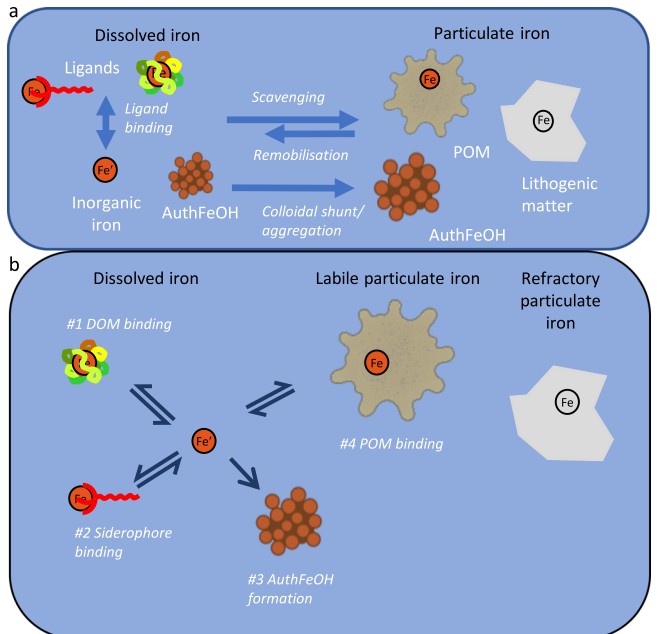

**Fig. 1 | Chemical processes that govern exchange between Fe pools in the ocean. a** Chemical controls are currently described as binding between inorganic Fe (Fe′) and ligands, scavenging onto particles, or precipitation as authigenic mineral Fe (authFeOH). Tagliabue et al.[3], introduced a further colloidal authFeOH shunt that moves Fe from the dissolved to the particulate phase via aggregation. **b** In this study we examine Fe partitioning between the dissolved and labile particulate phases, with refractory particulate Fe considered passive in the context of chemical interactions. We represent four different thermodynamically defined pathways comprised of #1 reversible Fe binding to dissolved organic matter (DOM), #2 reversible Fe binding to microbial siderophores, #3 Fe precipitation as authFeOH and #4 reversible Fe binding to particulate organic matter. We consider the organic phases to incorporate a wide range of Fe binding sites that reflect organic matter heterogeneity (see main text). Precipitated authFeOH could be present as colloids in the dissolved as well as in the particulate phase. All pathways operate at ambient pH and temperature. Further pathways, that would describe surface adsorption onto e.g., lithogenic particles are not included in this study.

Whilst the term scavenging implies a removal or loss of Fe, the process can be reversible, with potential abiotic remobilization of Fe from particles when particle concentrations increase as a result of hydrothermal or sedimentary inputs[17,28,29]. Iron chemically bound to POM is likely exchangeable, but the dynamics and role of exchangeable particulate Fe are not well understood, even though, in open ocean surface waters that do not receive lithogenic particles from the atmosphere, Fe is primarily associated with POM, and POM has a higher mass fraction than lithogenic matter[20].

Here, we examine if authFeOH formation and the mechanisms underpinning Fe losses from the ocean could be constrained within a thermodynamically consistent framework that accounts for organic matter binding site heterogeneity[9,30,31] within the context of changes in ambient pH and temperature through the ocean water column. We reframed the chemical controls on Fe as four reaction pathways that depend on Fe′ (Fig. 1b). Pathway #1 considers Fe binding to a heterogeneous DOM phase parameterised using the non-ideal competitive adsorption (NICA)-Donnan model[9,32]. Pathway #2 represents Fe binding to microbially produced siderophores[33], represented via stability constants that describe binding of the model siderophore ferrioxamine B with protons, major ions and Fe[34] and pathway #3 describes the formation of authFeOH[8]. In this study, we introduce a further pathway (#4) that represents a POM pool analogous to DOM, but with a separately defined distribution of binding sites. Ligand binding is therefore replaced by pathways #1 and #2, whilst reactions that lead to

authFeOH formation and abiotic reversible scavenging are represented by pathways #3 and #4, respectively.

We determined the impacts of these four pathways on calculated Fe′ concentrations and dissolved-particulate Fe partitioning along the zonal GEOTRACES GP21 transect in the South Pacific Ocean (SPO) at approximately 30°S. The SPO represents an ideal natural laboratory for this study given its low riverine and atmospheric inputs and well-defined water mass structure[35,36]. We restricted our analysis to waters deeper than 250 m to focus on depths where sinking particle fluxes are lowest[37] and biological processes involving photosynthesis, biogenic particle production, and remineralisation are likely less influential[26]. Organic particles are thus conceptualised as small (<53 μm) uncharacterised suspended particles[38,39] that make up the largest particulate pool by mass in the SPO[20] with a sufficient residence time within a water mass to reach equilibrium with Fe′[40,41]. We found that these four thermodynamically consistent pathways can produce distributions of dissolved and labile particulate Fe that closely match observations, with the exception of waters proximal to distinct Fe sources. Our results highlight the need for better constraints on the intrinsic binding properties of DOM and POM in biogeochemical models, as this is crucial for predicting how the ocean's Fe cycle will respond to future environmental changes.

## Results and discussion

### Physical processes have limited influence on iron distributions across the South Pacific Ocean

Observed DFe (DFe_obs, < 0.2 μm) concentrations varied from a minimum of 0.014 nmol L⁻¹ to a maximum of 12.6 nmol L⁻¹ across the SPO (Fig. 2a). Concentrations were elevated near the Saguaro Vent field on the East Pacific Rise (EPR, 117 °W; up to 12.6 nmol L⁻¹) and the Monowai Volcano (177°E; up to 3.7 nmol L⁻¹) in the west of the transect, reflecting hydrothermal and geological inputs. Concentrations up to 6.2 nmol L⁻¹ were also observed close to the South American shelf and slope at the eastern edge of the transect, as a result of Fe release from reducing sediments[28,42]. Beyond these source regions, DFe_obs concentrations were consistently higher at depths below 250 m east of the EPR (median (interquartile range (IQR)) = 1.05 (0.60) nmol L⁻¹) than west of the EPR (median (IQR) = 0.68 (0.43) nmol L⁻¹; Fig. 2a).

Observed total and labile particulate Fe (TPFe_obs and LPFe_obs respectively) concentrations (Supplementary Fig. 1a, Fig. 2b) were highest near to the Saguaro Vent field and the Monowai Volcano. Whilst the Hunga Tonga–Hunga Ha'apai eruption (December 2021) injected ash into the southwest Pacific shortly before our expedition (22 February to 08 April 2022)[43], we observed no evidence for a post-eruption ash signal at the depths we consider. Concentrations of TPFe_obs and LPFe_obs correlated with those of DFe_obs (LPFe_obs: $r = 0.74$, $p < 0.01$; TPFe_obs: r = 0.71, $p < 0.01$, $n = 403$) and were also slightly higher east of the EPR (median (IQR) LPFe = 0.086 (0.15) nmol L⁻¹; median (IQR) TPFe = 0.27 (0.62)) nmol L⁻¹) than west of the EPR (median (IQR) LPFe = 0.026 (0.043) nmol L⁻¹; median (IQR) TPFe = 0.103 (0.195) nmol L⁻¹). The enrichment in particulate Fe east of the EPR was influenced by South American shelf inputs[28,42], whilst any zonal propagation of the non-buoyant hydrothermal plume from the Saguaro Vent was likely constrained by the anticyclonic circulation around the EPR in this part of the SPO[44].

Water mass mixing has been shown to be important for distributions of dissolved cobalt, manganese and aluminium[45], which, like Fe[46–48], have oceanic residence times of less than a hundred years[6]. At depths below 250 m, the SPO is dominated by northward flowing waters from the Southern Ocean (Upper and Lower Circumpolar Deep Water, Antarctic Intermediate water: UCDW, LCDW and AAIW, respectively) and southward flowing Pacific Deep Water (PDW)[36]. Using end-member concentrations for these water masses (Supplementary Table 1), we determined the extent to which DFe distributions could be explained by water mass mixing via extended optimum

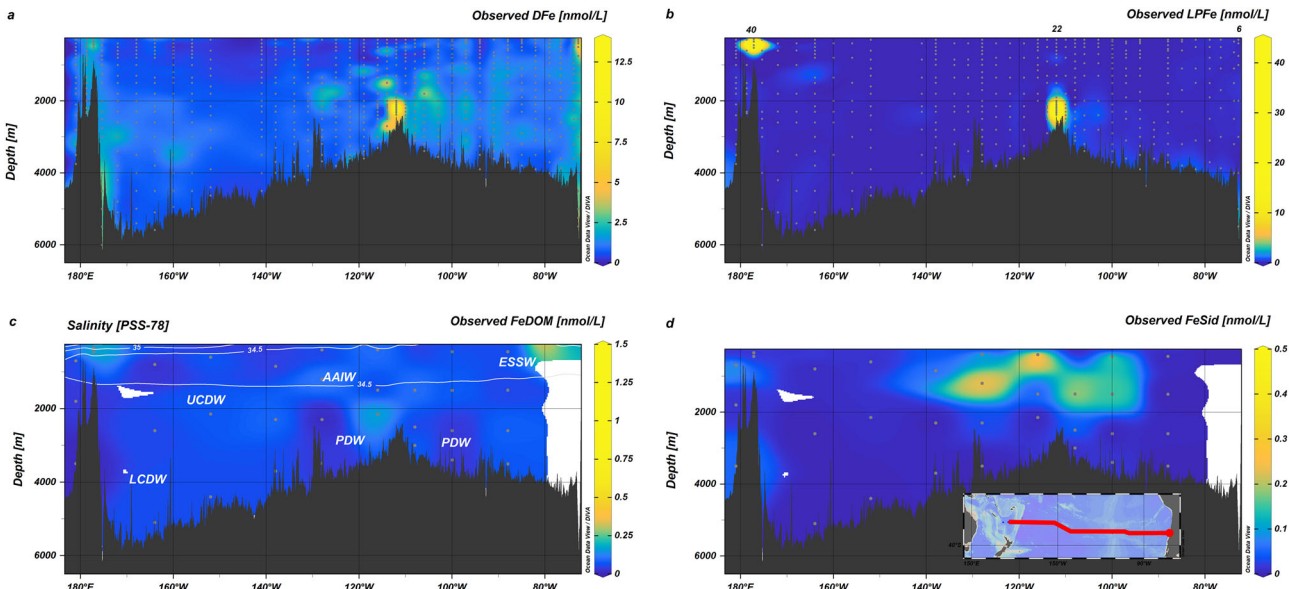

**Fig. 2 | Distributions of observed Fe fractions along a transect in the South Pacific Ocean. a** Dissolved Fe (DFe), **b** labile particulate Fe (LPFe), **c** iron bound to organic matter (FeDOM) with white contours showing salinity and **d** Fe bound to siderophores (FeSid). Inset shows map of the study area. Annotations in (**c**) show approximate locations of dominant water masses (ESSW Equatorial Subsurface

Water, AAIW Antarctic intermediate water, UCDW Upper circumpolar deep water, PDW Pacific deep water, LCDW Lower circumpolar deep water). Concentrations of FeDOM and FeSid are not corrected for analytical recovery (see Methods). Note non-linear colour scales. Images were generated in Ocean Data View.

---

multiparameter analysis[45]. We found that correlations between $DFe_{obs}$ and water mass composition were not significant (Supplementary Table 1). Conservative mixing between water masses does not explain the $DFe_{obs}$ distributions across the SPO, which is consistent with the high reactivity of Fe in seawater.

## Inorganic iron is supersaturated with respect to authigenic iron across the SPO

The weak influence of water mass mixing on $DFe_{obs}$ distributions, combined with the strong coupling of $DFe_{obs}$, $TPFe_{obs}$ and $LPFe_{obs}$ confirms that biological and chemical processes are key controls on Fe distributions[17,27]. Biological remineralisation of organic matter increases $DFe_{obs}$ concentrations in intermediate waters[27] and we observed a weak correlation between apparent oxygen utilisation and $DFe_{obs}$ concentrations in AAIW across the SPO (Supplementary Fig. 1b; $r = 0.35$, $p < 0.001$, $n = 93$) that reflects this process. Biological processes also influence pH and the distributions of organic matter, including siderophores, that determine the chemistry of Fe in seawater. Here, we explore the chemical controls on observed $DFe_{obs}$ inventories, focusing on the role of organic matter and authFeOH production (Fig. 1b, pathway #1 to #3 and Supplementary Fig. 2). Based on the ubiquitous presence of Fe bound to the chemically heterogeneous DOM fraction ($FeDOM_{obs}$)[10,11] determined along the GP21 transect (Fig. 2c), we first estimated equilibrium partitioning of Fe between hydroxo complexes and DOM (pathway #1, Supplementary Fig. 2a) using the NICA-Donnan model[9] and calculated FeDOM ($FeDOM_{pred}$) and Fe′ as a function of pH and DOM concentration (Supplementary Fig. 3) using a parameter set derived for marine DOM in the South Pacific[32]. In the NICA-Donnan model, Fe binding to DOM is governed by the total proton-binding site concentration, a bimodal distribution of site affinities representing DOM heterogeneity, and competition with protons and major ions for those sites[49]. Pathway #1 predicted Fe′ concentrations varied from a minimum of ~1–20 fmol L⁻¹ in the oligotrophic gyre of the western part of the transect to ~12 nmol L⁻¹ close to the Saguaro Vent field (Fig. 3). Inorganic Fe concentrations increased with increasing $DFe_{obs}$ but the relationship is non-linear (Fig. 4) as a result of DOM binding site heterogeneity.

In addition to the ubiquitous $FeDOM_{obs}$ pool, we observed a patch of relatively high concentrations of siderophores (up to 425 pmol L⁻¹) at depths between 400 and 1500 m in the SPO gyre (Fig. 2d) centred around 117°W, suggesting microbial Fe limitation[2] although outside of this patch siderophores could not be detected. Siderophores have a high affinity for Fe and will therefore compete with DOM binding sites when present. Our pathway #2 explicitly represents siderophores (Supplementary Fig. 2b)[33] in calculations of Fe′ at concentrations proscribed from interpolation of observed siderophore concentrations (Supplementary Fig. 3). The high affinity of the siderophore proxy (desferrioxamine B) means that the calculated concentrations of Fe bound to the siderophore ($FeSid_{pred}$) increased linearly with siderophore concentration for the most part (Supplementary Fig 4) which results in a reduction in predicted Fe′ when siderophores are present (Fig. 4). Nevertheless, Fe was still bound within the heterogeneous DOM pool even where siderophore concentrations are close to DFe concentrations (Fig. 4). Furthermore, use of desferrioxamine B as the proxy siderophore potentially overestimates FeSid concentrations since desferrioxamine B has a high affinity for Fe relative to the amphiphilic marine siderophores that dominate mesopelagic environments[2,50].

Near Fe sources, concentrations of Fe′ predicted from pathways #1 and #2 often exceeded the levels expected to trigger authFeOH formation[9,32] (orange line Fig. 4b). Inclusion of authFeOH formation at ambient $DFe_{obs}$ concentrations in our calculations (pathway #3, Supplementary Fig. 2c) caps Fe′ at concentrations equivalent to its solubility[9,32] (Figs. 3 and 4) as supersaturated Fe′ is predicted to precipitate as authFeOH. AuthFeOH likely first precipitates over timescales of days to weeks into colloids less than 0.2 µm in diameter, which would fall within the $DFe_{obs}$ size range[7,51], before aggregating into particles greater than 0.2 µm in diameter[3]. The most supersaturated waters were associated with the highest $DFe_{obs}$ concentrations close to the Saguaro Vent on the EPR[52] (Fig. 2), but supersaturation was also observed near the Monowai Volcano. Formation of authFeOH in proximity to hydrothermal systems is consistent with observations of authFeOH phases in particulate material

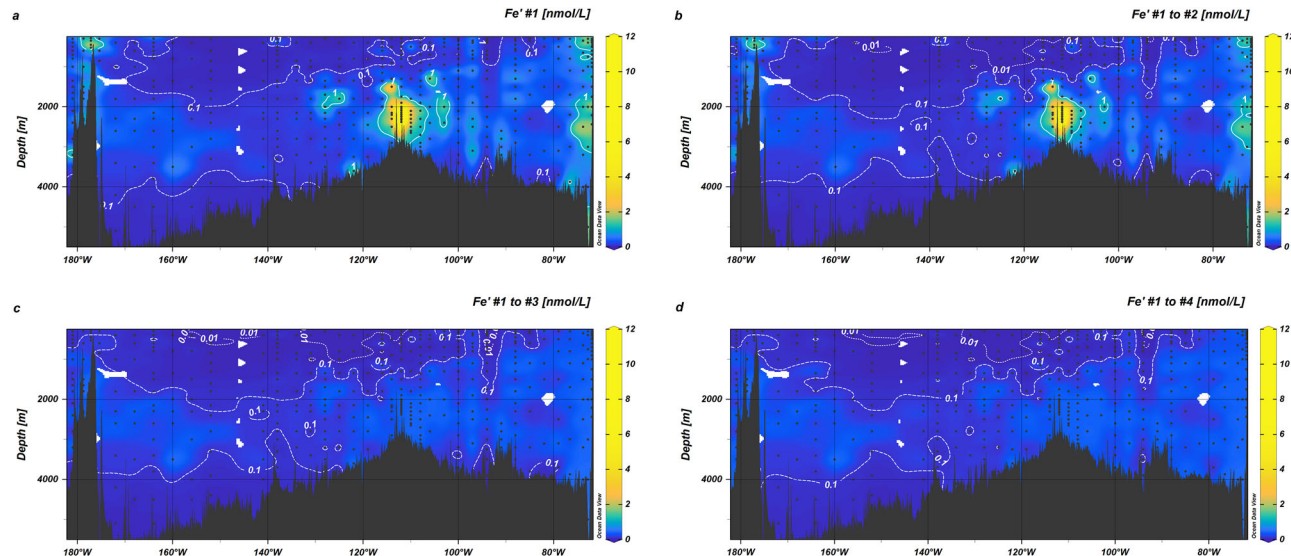

**Fig. 3 | Impact of reaction pathways on the spatial distribution of predicted inorganic Fe (Fe′) across the South Pacific Ocean. a** Pathway #1 (Fig. 1b) considered binding by dissolved organic matter (DOM). **b** Combined impact of DOM and siderophores on Fe′ (pathways #1 and #2). **c** Predicted Fe′ after incorporation of authFeOH formation (pathway #3). **d** Distribution of Fe′ across the SPO calculated after including POM (particulate organic matter) and labile particulate Fe (pathways #1 to #4; Fig. 1b). **a–c** Use dissolved Fe to define the total Fe pool, whilst (**d**) predicts Fe′ from all labile Fe phases (=dissolved + labile particulate Fe). Note non-linear colour bar scale. Contours indicate 1 nmol L$^{-1}$ (solid line), 0.1 nmol L$^{-1}$ (dashed line) and 0.01 nmol L$^{-1}$ (dotted line) boundaries.

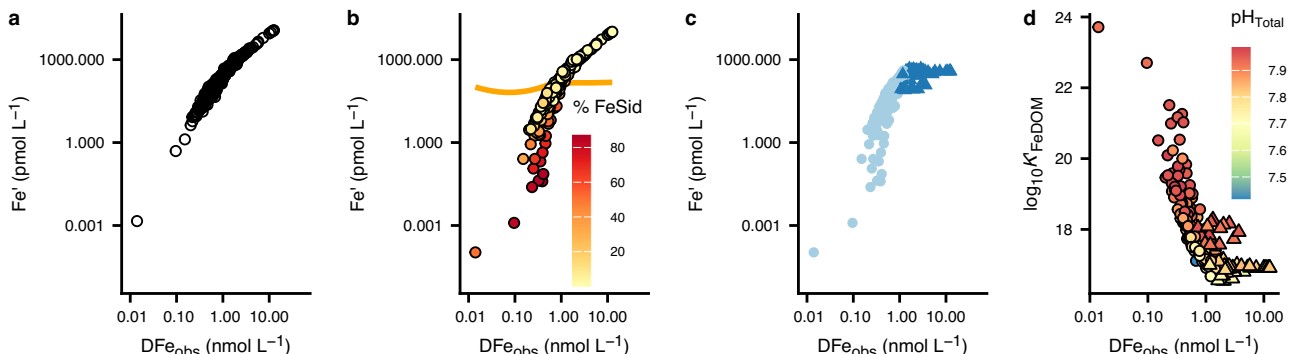

**Fig. 4 | Predicted changes in inorganic Fe (Fe′) and Fe-dissolved organic matter binding affinities (K′$_{FeDOM}$) as a function of observed dissolved Fe (DFe$_{obs}$).** **a** Relationship between predicted Fe′ and DFe$_{obs}$ when considering binding by dissolved organic matter (DOM; pathway #1, Fig. 1b). **b** Combined impact of DOM and siderophores on Fe′ as a function of DFe (pathways #1 and #2, Fig. 1b). The orange line represents the apparent solubility of authigenic Fe hydroxides (authFeOH) across the South Pacific Ocean, while colour shows the percentage of DFe bound to the siderophore. **c** Incorporating authFeOH formation (pathway #3, Fig. 1b) caps Fe′ below authFeOH solubility. Dark blue triangles indicate Fe′ where authFeOH is formed, and light blue circles show Fe′ where no authFeOH is formed. **d** Mean effective affinities of occupied DOM binding sites (K′$_{FeDOM}$) calculated when combining pathways #1 to #3 as a function of DFe concentrations. Triangles show effective affinities where authFeOH is formed and circles where no authFeOH is formed.

isolated close to hydrothermal sources[28,53] and the role of authFeOH formation as an important mechanism for Fe removal from hydrothermal plumes[54]. Iron supersaturation was also observed in proximity to the Peruvian shelf, likely linked to inputs of reduced Fe (Fe(II)) from sediments[32,42]. We note that redox processes are not accounted for in our model calculations.

We estimated the weighted mean effective affinities (Fig. 4) of occupied DOM binding sites from pathways #1 to #3 using the analytical solution to the NICA-Donnan binding affinity distribution[55]. We found the range of effective affinities for FeDOM at pH$_{tot}$ > 7.9 (16.5–23.7) was two orders of magnitude greater than the range of conditional stability constants for Fe binding ligands reported for the SPO (19.7–23.1 after adjustment to relate to free Fe$^{3+}$)[56] and six orders of magnitude greater than currently applied to describe ligand binding in global biogeochemical models[24,25]. We note that weighted mean effective affinities are not directly comparable to conditional stability constants. Weighted mean effective affinities of occupied sites under ambient conditions can be expected to vary more than conditional stability constants, which represent the mean affinity of all ligands that can be detected under given analytical conditions[4,57]. This wide range of effective affinities for occupied binding sites and their dependence on DFe$_{obs}$ (Fig. 4) reflects the impact of binding site heterogeneity, combined with the thermodynamic principle that binding sites are occupied in decreasing order of affinity. Minima in the effective affinities (Fig. 4) mark the threshold where DOM binding sites can no longer compete with authFeOH formation under the ambient conditions. It is accurate knowledge of this saturation state boundary at ambient DFe$_{obs}$, pH and temperature that determines the balance between Fe binding to organic matter and authFeOH formation. The saturation state boundary becomes more difficult to accurately determine if the binding site pool is averaged to a limited number of binding site groups with mean conditional affinities (Supplementary

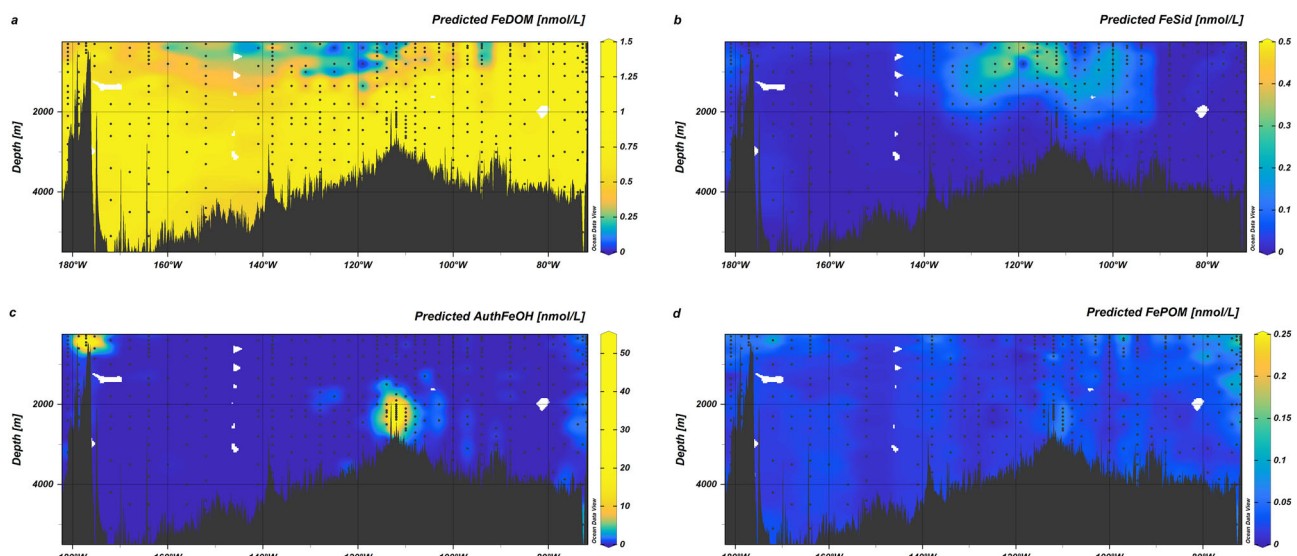

**Fig. 5 | Distributions of Fe phases predicted from combining interactions of total labile Fe (TLFe) with dissolved organic matter (DOM), a siderophore, formation of authigenic Fe (authFeOH) and particulate organic matter (POM).** **a** Iron bound to dissolved organic matter (FeDOM), **b** Fe bound to the siderophore (FeSid) (**c**) Fe precipitated as authFeOH, (note non-linear colour bar scale) and **d** Fe bound to particulate organic matter (FePOM).

Fig. 5). Hence, binding site heterogeneity is essential for reconciling binding of Fe to DOM with the role of authFeOH production as a key mechanism for Fe removal from the ocean[3,27].

**Particulate organic matter is an additional inorganic iron buffer**
Reversible exchange between dissolved, colloidal, and particulate Fe is required to explain long-distance transport of Fe plumes from hydrothermal systems[28] and elevated Fe concentrations associated with nepheloid layers[29]. Nevertheless, the main non-lithogenic component of the particulate Fe pool is typically assumed to be authFeOH[20,21,27] and the role of POM in Fe biogeochemistry is poorly understood. Both authFeOH and FePOM fractions contribute to the observed LPFe ($LPFe_{obs}$) fraction[58]. We therefore investigated interactions between Fe and POM, and authFeOH, via prediction of partitioning between DFe ($DFe_{pred}$) and the LPFe ($LPFe_{pred}$) fraction (pathway #1 to #4, Fig. 1b). We represented the functional group heterogeneity of POM, which remains largely uncharacterised[39], with the NICA-Donnan model. There is a paucity of investigations of cation binding to marine particles, but total binding site concentrations for marine particulate material have been estimated to be of the order of 1 mol binding site kg$^{-1}$ [59,60]. Accordingly, we modified generic parameters for humic substances[61,62] by setting the total binding site concentration to 1 mol binding site kg$^{-1}$ with a ratio of 3:1 between the low-affinity and high-affinity binding sites, in analogy to DOM[30]. We estimated the mass of POM particles from observed particulate total phosphorus using a total particulate carbon (C):phosphorus ratio of 80[37,63], which resulted in particulate organic C within the range (<0.01–0.68 µmol L$^{-1}$) observed for small particles (< 53 µm) across the SPO[37,64] (Supplementary Fig. 6a). We then assumed C comprised 50% of the POM mass[20,21] to obtain the amount of POM in kg L$^{-1}$ (Supplementary Fig. 6b). For inclusion of pathway #4, total Fe concentrations were set to the sum of $DFe_{obs}$ and $LPFe_{obs}$ (i.e., the total labile Fe concentration, $TLFe_{obs}$) and no prior assumptions were made with respect to partitioning between dissolved and particulate phases.

The $TLFe_{obs}$ concentrations in pathway #4 calculations were higher than the $DFe_{obs}$ values used in pathways #1 to #3, hence Fe′ was buffered to slightly higher levels than when only $DFe_{obs}$ was considered (median (IQR) = 164 (279) and 131 (246) pmol L$^{-1}$ respectively, $p < 0.01$, $n = 403$). The spatial distribution of Fe′ calculated considering

all four pathways showed increased concentrations in the eastern basin compared to the western basin (Fig. 3d; median (IQR) eastern basin 217 (286) pmol L$^{-1}$; median (IQR) western basin 84 (126) pmol L$^{-1}$), reflecting the overall pattern observed for $DFe_{obs}$ and $LPFe_{obs}$ across the SPO (Fig. 2). Iron binding to DOM was predicted to be widespread, but lower concentrations were calculated for the mesopelagic (250–1500 m) and where siderophore concentrations were higher (Fig. 5a). In the particulate phase, we calculated intense authFeOH production near the Saguaro Vent, Monowai Volcano and South American margin (Fig. 5c). Iron binding to POM ($FePOM_{pred}$) was calculated to be ubiquitous (Fig. 5d), with slightly higher abundance at depths towards the surface (< 500 m) and also at the South American margin, reflecting the distribution of POM (Supplementary Fig. 6b).

Occupied effective POM affinities (median (IQR) $\log_{10}K'_{FePOM} = 19.0$ (0.8)) were approximately 1.7 log units higher than those of occupied DOM sites (median (IQR) $\log_{10}K'_{FeDOM} = 17.3$ (1.0); Fig. 6). Higher affinities of occupied sites for the POM pool are required to explain partitioning of Fe into the particulate pool given the lower abundance of POM in comparison to DOM[65], but could also reflect preferential partitioning of more hydrophobic (e.g., more unsaturated) DOM components into the POM phase and an increase in the molecular weight of POM in comparison to DOM[39]. Effective affinities for both occupied $FePOM_{pred}$ and $FeDOM_{pred}$ binding sites decreased with depth (Fig. 6b). This reflects the heterogeneity of the organic matter pools: as the $TLFe_{obs}$ increases and organic matter decreases, progressively lower-affinity sites are occupied, lowering the weighted mean.

Overall, calculated Fe:C ratios for DOM ($Fe:C_{pred}$: median (IQR) = 18 (11) µmol mol$^{-1}$) compared well with observations ($Fe:C_{obs}$: median (IQR) = 14.4 (10) µmol mol$^{-1}$), whilst calculated Fe:C ratios for POM ($Fe:C_{pred}$: median (IQR) = 390 (150) µmol mol$^{-1}$) are in the upper range expected for non-lithogenic particles from the SPO (3–400 µmol mol$^{-1}$)[27]. Iron:$C_{pred}$ ratios were generally lower in the mesopelagic (< 1000 m) for both POM (median (IQR) = 290 (173) µmol mol$^{-1}$) and DOM (median (IQR) = 11 (9) µmol mol$^{-1}$) than in deeper waters (POM median (IQR) = 438 (100) µmol mol$^{-1}$; DOM median (IQR) = 21 (7) µmol mol$^{-1}$) (Fig. 6c). Preferential binding of Fe to the high affinity siderophore pool could be expected to reduce Fe:$C_{pred}$ within the OM pools and indeed the lowest calculated Fe:C ratios coincided with

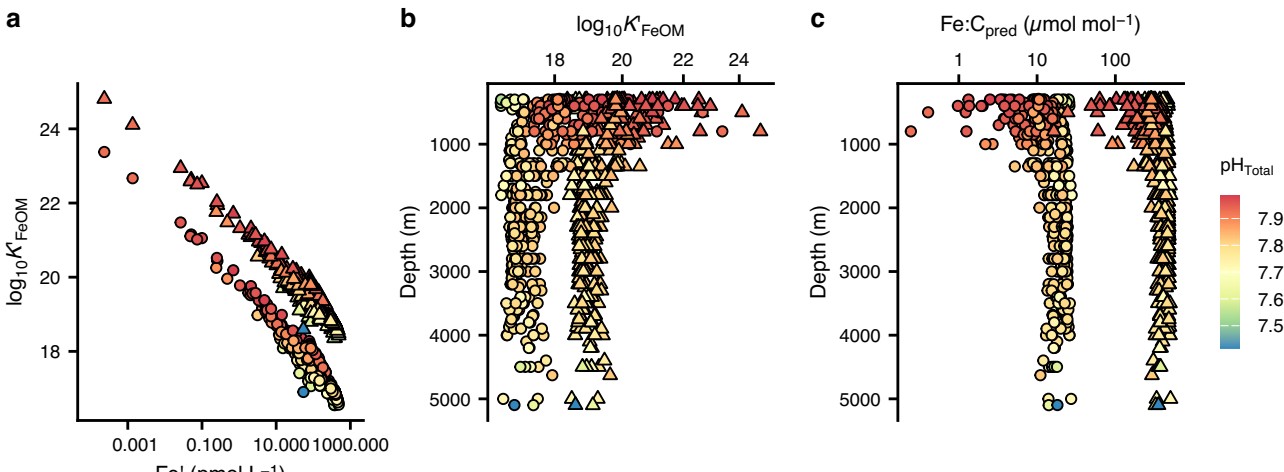

**Fig. 6 | Changes in predicted Fe-organic matter binding affinities and Fe:organic carbon ratios in the South Pacific Ocean.** Mean weighted effective affinities of occupied organic matter binding sites (logK'$_{FeOM}$) for Fe bound to particulate (triangles) and dissolved (circles) organic matter plotted as a function of (**a**) inorganic Fe (Fe′) and **b** water column depth. **c** Predicted Fe: carbon (C) ratios plotted as a function of water column depth. Colour indicates pH on the total scale.

highest siderophore concentrations. Nevertheless, calculations in the absence of siderophores (Supplementary Fig. 2e) confirmed that the trends in Fe:C ratios in DOM and POM were mainly driven by changes in the total Fe and C inventories through the water column.

We summed FeDOM$_{pred}$, FeSid$_{pred}$ and Fe′ to predict DFe concentrations (DFe$_{pred}$), and authFeOH and FePOM$_{pred}$ to predict LPFe$_{pred}$ concentrations. The major features in the DFe$_{pred}$ and LPFe$_{pred}$ distributions (Fig. 7a, b) reflected those of DFe$_{obs}$ and LPFe$_{obs}$ across the SPO (Fig. 2a, b). Predicted DFe correlated with DFe$_{obs}$ (Table 1), though the relationship became weaker at higher concentrations (Fig. 7c). Predicted LPFe showed a strong correlation with LPFe$_{obs}$ (Table 1), driven by authFeOH dominance at LPFe$_{obs}$ > ~1.5 nmol L$^{-1}$ (Fig. 7d). We observed a weaker correlation between LPFe$_{obs}$ and LPFe$_{pred}$ when FePOM was the only LPFe$_{pred}$ phase ($n = 113$, $p < 0.001$, $r = 0.2$), likely a result of uncertainties in the parameterization of FePOM. Although FePOM$_{pred}$ made up only ca. ~2% of the LPFe$_{pred}$ inventory, it was a critical component when considering the LPFe$_{pred}$ distribution (supplementary Fig. 7). Omission of POM binding from the calculations (Supplementary Fig. 2f) resulted in an absence of LPFe$_{pred}$ for ca. 65% of the datapoints since concentrations of authFeOH were not predicted to be saturating. Hence a widespread LPFe$_{pred}$ fraction could only be obtained when POM binding was included in the calculations.

Our predictions resulted in systematic variations between observed and predicted concentrations that depended on the Fe concentration. The spatial distribution of DFe and LPFe residuals (predicted-observed) showed that across much of the SPO, predicted and observed concentrations matched closely (median residuals 0.036 and −0.002 nmol L$^{-1}$ for DFe and LPFe respectively) (Fig. 7). We considered predicted concentrations within the interquartile range of LPFe residuals (inliers: residual LPFe > −0.034 or < 0.135 nmol L$^{-1}$, $n = 201$) to best approximate equilibrium conditions. Under these conditions, less than 10% of the LPFe datapoints were predicted to incorporate authFeOH. POM was thus the main carrier of Fe, where the exchange of Fe between solution and particles was governed by the concentration of Fe′, POM and the ambient pH. At equilibrium, the scavenging residence time of LPFe ($\tau_{LPFe}$) can be related to the residence time of particles ($\tau_{Part}$) via the equation[66]

$$\frac{\tau_{LPFe}}{\tau_{Part}} = 1 + \frac{c_{DFe}}{c_{LPFe}}, \tag{1}$$

where $c_{DFe}$ and $c_{LPFe}$ are the concentrations of DFe and LPFe respectively. Our calculations suggest the scavenging residence time of particulate Fe is approximately 36 times longer than the particles. A long FePOM residence time would then reflect a constant turnover of Fe between Fe′ and POM. Since the small POM fraction is sinking very slowly or is suspended[38,39], this pool of Fe could have very low to negligible loss rates, so that POM acts to further buffer Fe′ in the ocean.

## Dissolved and labile particulate iron anomalies are proximal to iron sources

Enhanced residuals (negative residuals DFe < −0.44 and LPFe < −0.315 nmol L$^{-1}$; positive residuals DFe > 0.453 and LPFe > 0.437 nmol L$^{-1}$) occurred mainly near hydrothermal, shelf and seafloor sources (Fig. 8). Given the proximity of these residuals to sources, we suggest they are caused by processes not considered in our model. We noted both negative and positive enhanced residuals for DFe and LPFe. Predicted DFe, which we calculated from summing FeDOM$_{pred}$, Fe′ and FeSid$_{pred}$, was lower than observed concentrations near the Saguaro Vent field, the Monowai Volcano (177°E) and the South American shelf and slope (Fig. 8a). Inputs from these sources are likely labile, dissolved and/or reduced forms of Fe (Fe(II))[28,42], and we suggest these anomalies reflect kinetic control of LPFe-DFe partitioning via a colloidal authFeOH phase (< 0.2 μm). Since we allocate authFeOH to LPFe$_{pred}$, our calculations do not reflect the presence of colloidal authFeOH, likely an important intermediary during authFeOH aggregation[3]. Overall, the rate of formation of authFeOH will be set by the slowest reaction that occurs as an Fe source mixes with ambient seawater. Reaction rates for interactions between metals and complex substrates at salinities, temperatures and pressures relevant to open ocean conditions are limited, and to our knowledge, no studies have specifically addressed the kinetics of Fe binding to POM. Dissociation of Fe from DOM has a half-life of less than a day at 20 °C[11]. Oxidation of Fe(II) can take hours to days under suboxic conditions[67], whilst authFeOH formation takes days to weeks at 25 °C[7,68]. Because most experimental studies were conducted at higher temperatures than the 1.6–16 °C range relevant here, the half-lives of these pathways are likely longer under in-situ conditions. Nonetheless, available evidence is consistent with authFeOH formation being the rate limiting step.

Near the seafloor, especially east of the EPR, DFe$_{pred}$ was higher while LPFe$_{pred}$ was lower compared to observed concentrations (Fig. 8). We observed elevated concentrations of labile particulate manganese (Mn, (LPMn)) coincident with negative DFe residuals at

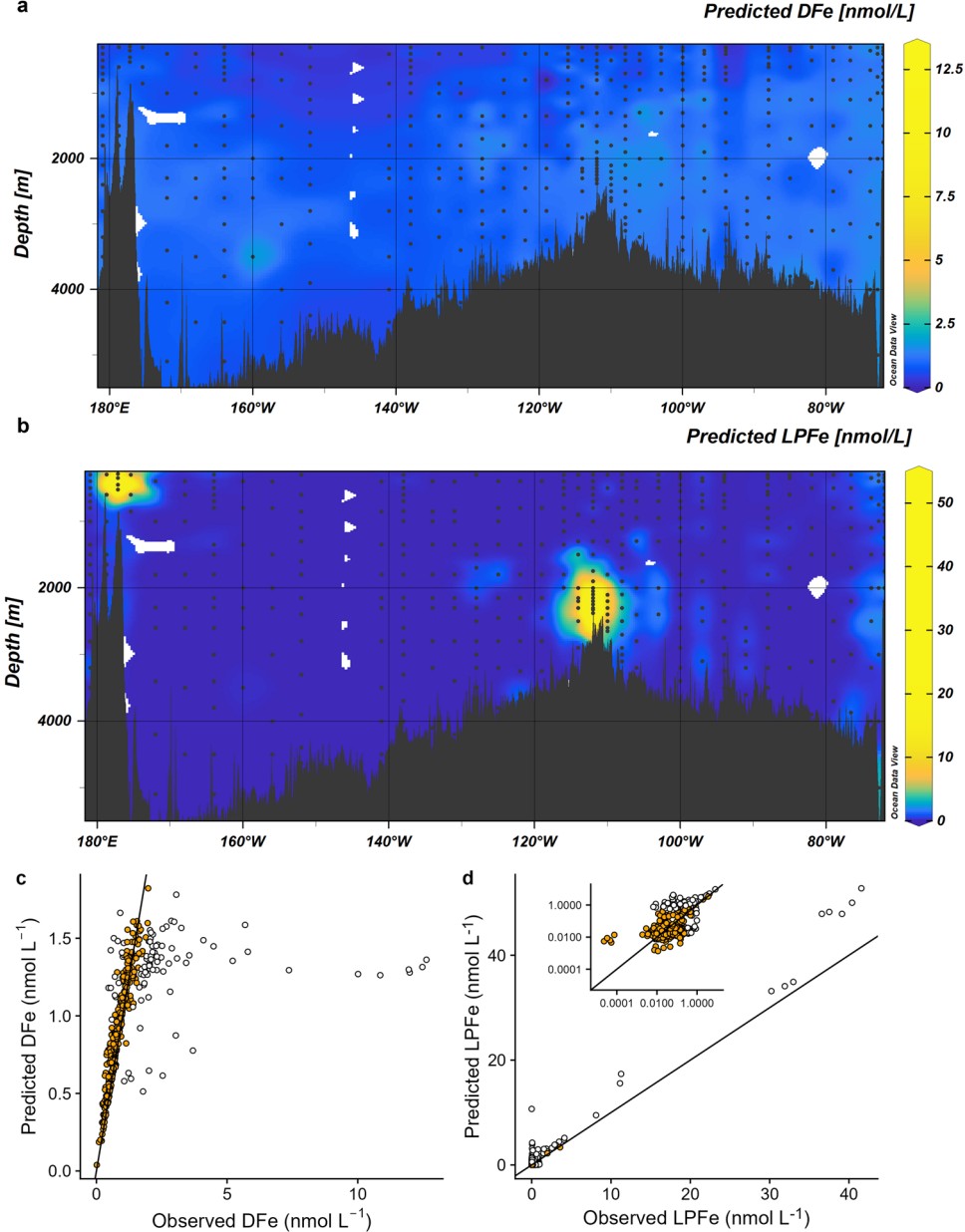

**Fig. 7 | Predicted dissolved and labile particulate Fe (DFe and LPFe respectively), and relationship to observed concentrations in the South Pacific Ocean.** Predicted (**a**) dissolved Fe and **b** labile particulate Fe across the South Pacific Ocean, **c** relationship between observed and predicted DFe concentrations and **d** relationship between observed and predicted LPFe concentrations. Values < 5 nmol L$^{-1}$ are shown in inset (note log scales on x and y axis). The black line in c and d shows the theoretical 1:1 relationship whilst orange points indicate inliers (i.e. residuals within the interquartile range (residual LPFe > −0.032 or < 0.157 nmol L$^{-1}$), which we considered best approximated equilibrium values.

depths greater than 3000 m (median (IQR) LPMn concentrations of 406 (173) pmol L$^{-1}$; $n = 11$). These elevate observed LPMn were ~20 times higher than observed LPMn concentrations (median (IQR) = 22 (14) pmol L$^{-1}$; $n = 201$) at depths where DFe is estimated to be at equilibrium with LPFe. The observed LPFe:LPMn ratios (1.1 ± 0.35 molFe:molMn) for data with negative LPFe residuals fell within the range of Fe:Mn ratios reported for Pacific ferromanganese formations[69]. We therefore consider the cause of enhanced negative residuals close to the seafloor to be a benthic source of ferromanganese oxides that is effectively inert under ambient seawater conditions[5]. Our equilibrium calculations therefore suggest that a benthic ferromanganese pool contributes to the LPFe$_{obs}$ pool, but because this Fe is likely present as aged oxyhydroxides that are

largely inert to seawater dissolution, they do not make a measurable contribution to DFe$_{obs}$ across the SPO.

## Implications and limitations
Our approach predicted equilibrium DFe and LPFe partitioning that was close to observed values across much of the SPO (Figs. 6, 7). The most critical pathways for constraining dissolved Fe phases were DOM (pathway #1) and authFeOH (pathway #3) since siderophores were constrained to the mesopelagic gyre and were thus absent in regions where Fe′ concentrations were sufficiently high to result in authFeOH formation. Accounting for the heterogeneity of organic matter binding site affinities was essential for a prediction of Fe′ concentrations consistent with authFeOH formation and accurate knowledge of the

**Table 1 | Summary of median observed and predicted DFe and LPFe concentrations**

| | | Dissolved Fe (nmol L⁻¹) | | Labile Particulate Fe (nmol L⁻¹) | |
|---|---|---|---|---|---|
| | | Observed | Predicted | Observed | Predicted |
| All data, n = 403 | Median (IQR) | 0.91 (0.74) | 1.01 (0.59) | 0.045 (0.123) | 0.027 (0.240) |
| | Linear regression | $DFe_{pred} = 0.10 \times DFe_{obs} + 0.85$, $r^2 = 0.20$, $p < 0.001$ | | $LPFe_{pred} = 1.2 \times LPFe_{obs} + 0.134$, $r^2 = 0.98$, $p < 0.001$ | |
| Data at equilibrium, n = 201 | Median (IQR) | 0.70 (0.44) | 0.73 (0.42) | 0.021 (0.022) | 0.019 (0.016) |
| | Linear regression | $DFe_{pred} = 0.94 \times DFe_{obs} + 0.12$, $r^2 = 0.92$, $p < 0.001$ | | $LPFe_{pred} = 1.1 \times LPFe_{obs} + 0.001$, $r^2 = 0.79$, $p < 0.001$ | |

Values are provided for all data and for data identified as best representing equilibrium, together with the equation for the linear relationship. Values in brackets show the inter-quartile range (IQR).

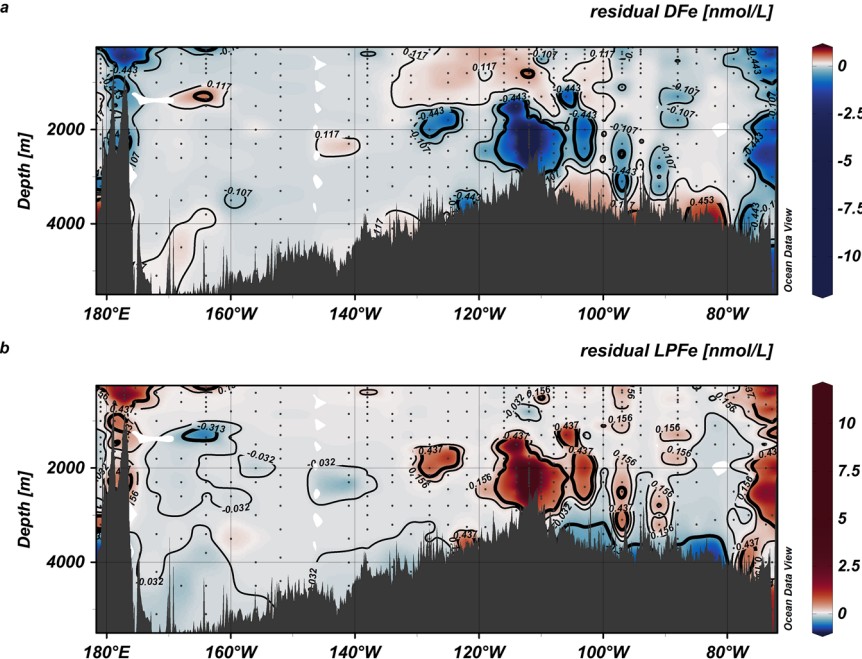

**Fig. 8 | Absolute residuals (predicted − observed) for dissolved and labile particulate Fe. a** Dissolved Fe (DFe) and **b** labile particulate Fe (LPFe) across the South Pacific Ocean. Thin contour lines indicate the boundary for observations estimated to be at equilibrium based on the interquartile ranges ($q_{25}$ DFe = −0.107 and LPFe = −0.032 nmol L⁻¹; $q_{75}$ DFe = 0.117 and LPFe = 0.157 nmol L⁻¹), whilst thicker lines mark the boundary for enhanced residuals (negative residuals DFe < −0.44 and LPFe < −0.315 nmol L⁻¹; positive residuals DFe > 0.453 and LPFe > 0.437 nmol L⁻¹).

minimum affinity for binding sites that can be occupied under ambient conditions is necessary to constrain authFeOH formation. Hence it is particularly important to understand the role of binding site heterogeneity in Fe biogeochemistry and further investigations into how this varies with DOM abundance and lability[30,32] are required. Precipitation of authFeOH depends strongly on pH and temperature: Fe′ solubility increases as both temperature and pH decrease, deepening the authFeOH saturation horizon in the ocean[9]. Conversely, the higher pH and temperature in lower latitude surface waters would reduce Fe solubility, which could impact on the overall efficacy of Fe inputs to the surface ocean in these regions.

Addition of a ubiquitous POM phase that could equilibrate with the total labile Fe pool (pathway #4) allowed us to deconvolute the particulate Fe phase into a ubiquitous FePOM phase present at low concentrations, and an authFeOH phase that precipitates over the course of days to weeks when Fe′ concentrations become saturating. The POM phase represents reversible scavenging and its response to changes in pH, temperature, TLFe concentration and POM abundance. We restricted our analysis to depths below those expected to be strongly influenced by biological processes[37]. Hence, in surface waters or the upper mesopelagic, where microbial remineralisation processes are intense, increased microbial activity could also influence Fe

retention within the POM pool[26,70]. Examination of residual predicted-observed concentrations also allowed us to identify an inert ferromanganese phase with a benthic source (Fig. 9).

Our findings suggest that binding of Fe to POM can explain the ubiquitous presence of a low abundance LPFe pool. However, FePOM interactions remain the most uncertain part of our calculations, and better knowledge of the binding properties, binding dynamics and functional group composition of suspended POM[38,39] is required to reduce this uncertainty. We also did not include interactions with other types of particulate surfaces, such as biogenic carbonates or silicates[5,19,26,64]. These phases likely represent only a minor fraction of the total particulate pool in this study area[64], and since metal affinities for surface adsorption sites on these phases are generally lower than for POM[66], surface adsorption could be a minor factor in Fe biogeochemistry across the SPO. Nevertheless, this supposition requires further, independent verification.

Around the Saguaro Vent where authFeOH formation is pronounced, particulate Fe dominates the total Fe pool (DFe < LPFe) and the residence time for LPFe will be close to that of the authigenic particles themselves. In contrast, under equilibrium conditions, where POM plays a more dominant role and LPFe is less abundant than DFe (Table 1), the scavenging residence time of LPFe will be longer than

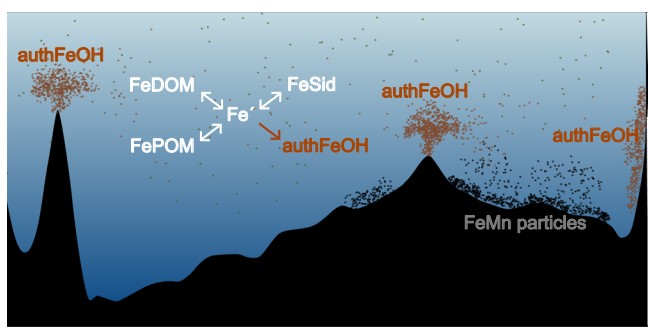

**Fig. 9 | Chemical controls on Fe across the South Pacific Ocean (SPO).** Equilibrium processes (white text) dominate in the oceans' interior. At equilibrium, concentrations of inorganic Fe (Fe') are buffered by Fe binding sites within dissolved (DOM) and particulate organic matter (POM) pools, with siderophores (Sid) contributing when they are present at high enough concentrations. When Fe' becomes saturated, authigenic Fe hydroxides (authFeOH) form, which can be lost from the water column or, if present at low concentrations, remain within the water column, potentially as colloids and in association with organic matter. At the margins, equilibrium conditions are disturbed by Fe inputs. Inputs of labile and reduced Fe from e.g., hydrothermal, volcanic or reducing sediments give rise to supersaturated Fe' that precipitates as authFeOH at rates likely controlled by a combination of pH, temperature and the form of Fe supplied by the source. The high abundance of Fe within this authFeOH phase means that rates of Fe loss will be intense near Fe sources. Our approach suggests inert sources of Fe, e.g., in the form of aged ferromanganese particles or refractory lithogenic material, do not result in detectable increases in dissolved forms of Fe in SPO.

that of the residence time of particles. Thus, Fe losses in the SPO are likely mainly driven by authFeOH removal, with the greatest losses occurring near Fe sources. Our results therefore, support the role of authFeOH formation as a key controlling process regulating DFe concentrations in the ocean[3,27].

We have reframed the chemical controls on Fe across the SPO by predicting its binding to DOM, siderophores and POM, as well as its precipitation as authFeOH. We found that incorporating binding-site heterogeneity in DOM and POM, together with ambient variations in physicochemical conditions, produced predicted distributions of particulate and dissolved Fe that closely matched observations. Accounting for binding site heterogeneity is critical for predicting authFeOH formation within a thermodynamically consistent framework. Discrepancies in predicted Fe were found near the Saguaro Vent system (EPR), the Monowai Volcano (west of the transect), and the South American margin (east of the transect), which we attribute to kinetic control of authFeOH formation, likely via a colloidal phase. We also identified a benthic input of LPFe east of the EPR, attributed to a supply of ferromanganese particles that are relatively inert under the oxic, slightly basic conditions of the deep ocean. Our results confirm that authFeOH formation is the primary mechanism driving dissolved Fe losses below the surface ocean. Our approach suggests POM acts a carrier for Fe that will partition between dissolved and particulate phases according to the concentrations of Fe', POM and ambient pH. Hence for Fe, abiotic scavenging mainly occurs via loss of particles in which Fe itself is the dominant substrate. However, we highlight that there are considerable uncertainties in our parameterisation of the POM pool, and independent determination of binding and kinetic properties of particulate materials as a function of temperature and pH would better constrain these interactions. Since the chemical processes that control Fe also affect other metals, further efforts to move beyond the black boxes of ligand binding and scavenging would likely reframe our understanding of trace element biogeochemistry in the ocean.

## Methods
### Analysis of dissolved iron and particulate iron and phosphorous
Seawater samples were collected on a transect of the South Pacific Ocean (SPO) from Valparaiso (Chile) to Noumea (New Caledonia) (German GEOTRACES cruise SO289/GP21; Fig. 2), on the FS Sonne following GEOTRACES protocols[71]. Dissolved and particulate trace metals were collected using a titanium rosette equipped with 24 trace metal clean 12 L externally sprung Niskin bottles (Ocean Test Equipment) and a Seabird 911 conductivity-temperature-density (CTD) sensor. Dissolved trace element samples were filtered (0.2 μm, AcroPak 500, Pall) into 125 mL low density polyethylene (LDPE) acid-cleaned bottles (Nalgene). All seawater samples were acidified with 12 M hydrochloric acid (Ultrapure, Romil) in a laminar flow bench (pH ~1.9). Particulate trace metals and phosphorus were pressure filtered (N$_2$ gas at ~0.5 bar) onto acid-cleaned 25 mm 0.45 μm pore-sized polyethersulfone (PES) membrane filters. Approximately 2–8 L passed through each filter depending on depth and region. Filters were stored frozen (−20 °C) prior to processing and analysis at GEOMAR.

Dissolved iron (DFe) was determined after preconcentration and matrix removal using an automated SeaFast (SC-4 DX; ESI) fitted with a Nobias resin column[72] and analysed by high resolution inductively-coupled mass spectrometry (HR-ICP-MS; Element XR, ThermoFisher). All reagents were prepared in deionized water (MilliQ; 18.2 MΩ.cm). Nitric acid was purified by single distillation in a sub-boiling per-fluoroalkoxy-polymer (PFA) distillation unit (DST-1000, Savillex). Samples, standards and reference waters were UV oxidized (Osram PURITEC; 25 watts; >5 h) prior to analysis. Weighted mean absolute percentage errors, calculated from analysed SAFE, CASS and NASS reference materials were < 6.5% for DFe. Relative standard deviations (RSD) of selected seawater samples analysed in triplicate were < 7.1%. Limits of detection (LOD), calculated for each HR-ICP-MS run, ranged from 0.02 to 0.12 nmol L$^{-1}$.

Particulate Fe, phosphorus (P) and manganese (Mn) concentrations were determined after sequentially leaching and digesting the PES filters into labile and refractory fractions[22,58]. The labile fraction (LP) was determined after leaching in 25% (v/v) acetic acid (Optima grade, Fisher) and 0.02 M hydroxylamine hydrochloride (Sigma) for 2 h, with a short heating step at the beginning (95 °C for 10 min). Residual particulate material was then digested under reflux (150 °C for 14 h) in a solution of 50% HNO$_3$ and 10% HF (v/v; Optima grade, Fisher). For both fractions, the leaching acid was evaporated (~2.5 h at 110 °C) and the residual oxidized by adding 0.5 ml of a 50% HNO$_3$ (v/v) and 15% H$_2$O$_2$ (v/v) solution to each vessel for 30 min. The oxidising solution was removed by evaporation (1 h at 110 °C) and LP and residual fractions were dissolved into 4.5 ml of 1 M HNO$_3$ and stored in acid washed 15 ml centrifuge tubes. Digested samples were analysed by HR-ICP-MS. Average LODs for particulate analysis were 0.82, 1.89 and 0.0073 pmol/filter for Fe, P and Mn respectively. Average RSDs were <4%, <12% and < 5% for Fe, P and Mn.

Concentrations were blank corrected and verified using BCR-414 certified values. Recoveries for BCR-414 were: 107.5 ± 16.4% for Fe, 97.6 ± 28.4% for P and 99.6 ± 18.8% for Mn. Filter blank averages were: ~43, ~102 and ~0.18 pmol/filter for Fe, P and Mn.

### Analysis of dissolved organic carbon and pH
Samples for analysis of dissolved organic carbon (DOC), dissolved inorganic carbon (DIC) and total alkalinity (TA, used to calculate pH) were collected at the same stations as trace metals using a conventional stainless-steel rosette equipped with 24 internally sprung Niskin bottles (Ocean Test Equipment). Dissolved organic carbon samples were filtered through pre-ashed glass fiber filters (0.7 μm) into combusted glass vials, acidified to pH 2 with reagent grade hydrochloric acid and then flame sealed. Total dissolved organic carbon (DOC) was quantified after high temperature catalytic oxidation (Shimadzu TOCN

analyser)[73]. Outliers (randomly scattered high values) at depths > 250 m were identified as data points that were over the upper limit of the 95% confidence interval calculated for the whole transect (>52.4 µmol L$^{-1}$) and were not considered in the interpolation. The mean DOC concentration determined for Miami Mid depth consensus reference material #11–21 was 59.1 ± 4.1 µmol L$^{-1}$, which compares to the consensus range of 55–57 µmol L$^{-1}$.

Samples for TA and DIC were collected into wide neck 250 mL borosilicate glass bottles and poisoned with mercuric chloride[74]. Total alkalinity and DIC were determined using a Versatile Instrument for the Determination of Total inorganic carbon and titration Alkalinity (VINDTA 3 C, Marianda, Germany), calibrated using certified reference material (Dickson, Scripps Institution of Oceanography, USA)[75]. pH values on both total and NBS scales were calculated with PyCO2SYS[76].

### Analysis of dissolved iron bound to siderophores and organic matter

Samples for siderophores and dissolved organic matter (DOM) were collected using the titanium rosette at approximately every third station across the transect (Fig. 2c, d). Unfiltered seawater was siphoned directly into methanol rinsed and acid washed (0.01 M HCl) reagent bags (2 L, Flexboy, Sartorius). Samples were gravity filtered in the dark through a polyvinyl difluoride cartridge membrane filter (0.2 µm, Sterivex, Sartorius) and then over a solid phase extraction cartridge (500 mg, ENV + , Agilent) that had been preconditioned with methanol and cleaned with 0.01 M HCl[10,77,78]. Cartridges were air dried and stored frozen (-20 °C).

Prior to analysis, cartridges were rinsed in 11 mmol L$^{-1}$ ammonium carbonate (pH 7.8) and retained organic matter, including siderophores, eluted with 5 mL of a mixture of H$_2$O:isopropanol:acetonitrile (5:15:80 v:v:v)[78]. A 1 mL aliquot was then reduced under vacuum to approximately 100 µL before analysis. Siderophores and DOM were separated by high performance liquid chromatography (HPLC, biocompatible Ultimate 3000, Thermo) over a polymeric PEEK column (Hamilton PRP, 2.1 × 150 mm, 5 µm). Iron associated with siderophores and DOM was quantified by HR-ICP-MS operated in-line with the HPLC after desolvation (Aridus II, CETAC)[10,33] to remove organic solvents. Recoveries for siderophores were estimated to be 82 ± 24% based on extraction of ferrioxamine B, ferrichrome and ferrioxamine E spiked into siderophore free seawater (n = 9). Dissolved organic carbon in the extracts was determined after high temperature catalytic oxidation after removal of solvents by evaporation at 40 °C overnight and redissolution of the residual in 0.1 M HCl[10]. Dissolved organic carbon recoveries were determined to be 11 ± 4%. Recoveries of FeDOM were thus likely to be considerably lower than those of siderophores. Concentrations of FeDOM and siderophores were not corrected for recoveries.

Iron siderophore and DOM peaks were deconvoluted by fitting exponentially-modified Gaussian[79] peaks in combination with a linear estimation of baseline drift via a Levenberg-Marquardt fitting routine[80]. Iron was quantified using ferrioxamine B as the standard. The concentration of Fe bound to ferrioxamine B was verified by off-line comparison of the ferrioxamine B standard with ICP-MS standards (Inorganic Ventures).

The relative analytical uncertainties for Fe bound to DOM and siderophores was calculated to be 27% based on repeated analysis of ferrioxamine B (n = 9) over the course of the sample analysis. The DOM and siderophore peaks were identified based on their molecular and spectrophotometric properties, which were collected after splitting the HPLC outflow to a diode array detector (Ultimate 3000 DAD, Thermo) and a high-resolution electrospray ionisation mass spectrometer (HR-ESI-MS, Q Exactive, Thermo) that was operated in parallel to the HR-ICP-MS[10]. Flow rates were checked with a liquid flow meter (GJC Instruments) to ensure the split was maintained at a constant level (1:2 HR-ICP-MS: HR-ESI-MS) throughout analysis. DOM peaks were broad,

and demonstrated complex ESI-MS mass spectra and strong absorption at 254 nm, while siderophore peaks were narrower and could be attributed to amphibactin, marinobactin and ferrioxamine siderophore classes[10,33]. We report the sum of all siderophores determined as their Fe complexes in this manuscript.

### Ancillary parameters

Macronutrients nitrate, nitrite (NO$_3$, NO$_2$), phosphate (PO$_4$) and silicate (SiO$_4$) were analysed on board using an auto analyzer (QUAATRO39, Seal Analytical)[81]. Seawater reference samples (Kanso, Japan) were used for data quality control. Depths, salinity, oxygen and temperature were obtained using Seabird CTD systems fitted to the rosette frames. Salinity was derived from conductivity after calibration against discrete samples analysed with a salinometer (Guildline) and the oxygen sensor was calibrated using Winkler titrations[81].

### Extended optimum multiparameter analysis

Water mass mixing was determined via extended optimum multiparameter analysis (eOMPA)[82]. Parameters and results are given in Table S1. Mixing was assessed based on source water parameters of temperature, salinity, oxygen (O$_2$), and major nutrients of 6 major water masses in the SPO[83]. Endmembers for the major water masses were adapted from Talley et al.[83] (Table S1). Redfield ratio parameters used for eOMPA were: 0 (temperature), 0 (salinity), -170 (oxygen), 1 (phosphate), 16 (total nitrate + nitrite), 40 (silicate).

### Compilation of the data set used for calculations

We based our data set on Fe observation at depths > 250 m (n = 403). Dissolved organic carbon concentrations were obtained on a separate CTD cast hence depths did not always overlap with those of the metals, whilst pH values and siderophore concentrations did not have the same coverage as the Fe data. We therefore linearly interpolated between data points to a grid of 10 km by 1 m in R using the akima package[84] to match pH, DOC and siderophore concentration to Fe data between 72°W and 179°E. For pH, issues with the instrumentation meant that only a limited data set was available from the cruise itself, with better coverage towards the western part of the transect. We therefore used data from the 2017 P06 Clivar transect available in the GLODAPv2022 database (Cruise ID 320620170820)[85] as a base for interpolation of pH up to 146°W, and data analysed from samples collected on the cruise for interpolation of pH between 146°W and 179°W[85]. Our interpolated pH (Supplementary Fig. 3b) was in good agreement with trends observed for pH across the South Pacific Ocean[9,86]. The mass of POM particles was estimated from observed total particulate phosphorus (TPP) using a particulate organic carbon: TPP ratio of 80 obtained for the small particle fraction collected across the SPO at 16°S[37,63,87]. Calculated particulate organic carbon (POC) was confirmed to be within the range (< 0.01–0.68 µmol L$^{-1}$) observed for small particles (< 53 µm) across the SPO[37,64] (Supplementary Fig. 6a). We observed elevated values of TPP (21-44 nmol L$^{-1}$) between 2050 and 2380 m in proximity to the Saguaro Hydrothermal Vent field (Station 22, results not shown), similar to trends observed on the GP16 cruise that crossed the EPR at 16°N[37,64]. Since, no elevations in POC were observed close to the EPR on GP16[37,64] (Supplementary Fig. 6a) we suspected these elevated LPP values were not organic in origin. Hence, we set the POM concentrations to the median POM (2 × 10$^{-9}$ kg L$^{-1}$) between 2050 and 2380 m at Station 22. We then assumed carbon comprised 50% of the POM mass[20,21] to obtain the amount of POM in kg L$^{-1}$ (Supplementary Fig. 6b).

### Chemical equilibrium calculations for simultaneous prediction of multiple Fe species

Equilibrium calculations were carried out using the speciation program ORCHESTRA[88]. For each sample, the model takes measured

environmental conditions (major ions, pH, T and DOM/POM) together with a total Fe boundary condition ($DFe_{obs}$ for pathways #1-#3, $TLFe_{obs}$ for pathway #4) and returns the equilibrium distribution of Fe among the defined pools (Fe', FeDOM, FeSid, authFeOH and FePOM), from which $DFe_{pred}$ and $LPFe_{pred}$ are computed (Supplementary Fig. 2). The parameters and input files used for this analysis together with instructions for making calculations in ORCHESTRA are available online. Binding to DOM (pathway #1, Fig. 1b & Supplementary Fig. 2a) is calculated according to the NICA-Donnan model[89], using Fe binding parameters obtained for South Pacific waters[32]. Our use of one parameter set for the whole transect is supported by minimal variability in DOM composition across the SPO, especially at depths > 250 m[90]. We used a conversion factor of 0.0408 (kg DOM) (mol$^{-1}$ DOC) to obtain DOM masses for the calculations (Supplementary Fig. 3d). A siderophore proxy (pathway #2, Supplementary Fig. 2b) was added to the model via addition of stability constants for ferrioxamine B binding to protons, magnesium, calcium and Fe[33,34]. Although this does not represent the true heterogeneity of the siderophore pool[2,33], it allowed us to incorporate a separate type of Fe binding ligand that is independent of DOC concentrations. The temperature dependent parameterisation of Fe hydroxides and ferrihydrite precipitation within ORCHESTRA (pathway #3, Supplementary Fig. 2c) was updated to be compatible with the solubility constants derived for sodium chloride[7,32]. Ferrihydrite precipitation was considered a proxy for authigenic Fe mineral (authFeOH) formation according to the equation[7]

$$Fe^{3+}(aq) + 3H_2O \rightarrow Fe(OH)_3(s) + 3H^+(aq) \tag{2}$$

hence in our model (pathway #3, Fig. 1b & Supplementary Fig. 2c), OM determines the concentration of $Fe^{3+}$, which precipitates when the solubility of $Fe^{3+}$ (which is linearly related to concentrations of Fe') is exceeded. In pathways #1-#3, we used DFe (< 0.2 μm) as the input Fe concentration and predicted authFeOH could therefore be in the colloidal phase since the parameters on which authFeOH formation are based considered authFeOH to be > 0.02 μm[7]. Nevertheless, close to sources of Fe, it is also possible that the system is supersaturated with Fe' and has not yet reached equilibrium. For pathway #4 (Fig. 1b & Supplementary Fig. 2d–f), the total Fe concentration was assumed to be the sum of DFe and LPFe. Particulate organic matter was represented by a particulate NICA-Donnan phase with two binding site groups – a low affinity group (POM1) and a high affinity group (POM2), with parameters based on those of the generic humic acids[61,62]. We reduced the total amount of binding site groups to 1 mol binding site kg$^{-1}$ POM[59,60] and set the ratio of group 1: group 2 binding sites to 3:1 to reflect the ratio observed for marine DOM[30,91,92]. The mass of POM (kg L$^{-1}$) was calculated from total particulate phosphorous concentrations using a C:P ratio of 80[37,63,87] and assuming C is 50% of the POM mass[20,21]. Although we examined the sensitivity of our results to Fe parameters and attempted to optimise the FePOM binding parameters using PEST-ORCHESTRA[32], we could not achieve an improvement on our initial results. We highlight the hypothetical nature of these parameters and stress that independent experimental evidence is required to derive parameters describing FePOM interactions. Further details on Fe speciation calculations are provided in the Supplementary Information.

## Data availability
The data generated in this study have been deposited in Zenodo under accession code https://doi.org/10.5281/zenodo.17456230.

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

## Acknowledgements

The authors would like to thank the captain and crew of the RV Sonne for their help and support. We thank L. Blum for assistance with sampling, A. Mutzberg for nutrient analysis and T. Steffens for his assistance with the trace metal analysis. This work was funded by the German Federal Ministry of Research, Technology and Space (Grant number: 03GPF 18-1_032) and by the Helmholtz Association.

## Author contributions

M.G.: Design, conceptualization, analysis and writing – original draft preparation; K.G., M.P.H., L.D., N.H.: analysis and writing – review and editing; K.Z.: software; P.L., C.R.C.: writing – review and editing; E.P.A.: Funding acquisition, conceptualization and writing – review and editing.

## Funding

## Competing interests

The authors declare no competing interests.
