## [Peer Review file · Nature Communications]

Chemical controls on iron distributions across the subsurface South Pacific Ocean

Corresponding Author: Dr Martha Gledhill

Version 0:

Reviewer comments:

Reviewer #1

(Remarks to the Author)

Outstanding manuscript and certainly "noteworthy"!

Direct answers to reviewer feedback prompts:

1. What are the noteworthy results? Gledhill et al. have synthesized a substantial data set of superior quality to produce a manuscript interpreting multiple interwoven processes into a cohesive description of Fe cycling in the South Pacific Ocean. Just the amount of analytical work required to produce the different data sets is herculean in scope and detail, but to then interpret all the concentrations, processes, and biogeochemical mechanisms in context...Nature Communications exists for publishing manuscripts like this!

2a. Will the work be of significance to the field and related fields? Yes! The interpretations and conclusions described in this manuscript help to conceptualize the major processes describing Fe biogeochemical cycling, with predictive power on Fe limitation and supply in diverse oceanographic regimes. In addition, the authors' approach to synthesizing multiple data sets provides a framework for other marine biogeochemists to interpret past and future data sets in the proper context of multiple related parameters.

2b. How does it compare to the established literature? The authors provide sufficient QA/QC values for all their parameters to allow direct comparison with past and future studies of one or more Fe data sets. Furthermore, the authors demonstrate the relevance of their research to the broader studies of biological and chemical oceanography, citing relevant publications that will allow future studies to discuss individual or multiple Fe data sets in the context of the "big picture."

3. Does the work support the conclusions and claims, or is additional evidence needed? The authors have provided EXCEPTIONAL evidence in support of their conclusions and claims. Seriously, each data set could be published individually and be noteworthy...seeing all these data sets and supporting data synthesized into a unified interpretation is beyond exemplary.

4. Are there any flaws in the data analysis, interpretation and conclusions? Do these prohibit publication or require revision? The authors sufficiently discuss any weaknesses in their synthesis, especially with respect to surface adsorption onto specific biominerals like silicate frustules or calcite tests. Instead of considering these acknowledgements as "flaws," they are properly noted as areas for future investigations, and do not in any way prohibit publication.

5. Is the methodology sound? Does the work meet the expected standards in your field? The methodologies and supporting QA/QC are exceptional, and only in a more focused publication on a single parameter would I expect to see additional analytical figures of merit.

6. Is there enough detail provided in the methods for the work to be reproduced? Absolutely! The authors give sufficient evidence in the supplemental materials for their own data quality as well as relevant references for their work to be duplicated.

I have only a few minor suggested revisions before publication, but I trust the editor(s) and authors to handle these minor updates without further input from me.

Lines 272-273: statement is a little awkward to understand. Do the authors mean that the solution and particles were governed by the concentration of Fe', POM, and ambient pH? Perhaps revise to read as:
"POM was thus the main carrier of Fe, where the exchange of Fe between solution and particles was governed by the concentration of Fe', POM, and the ambient pH"

Line 336: change "deconvoluted" to "deconvolute"

Line 336: did you mean to write "FePOM" instead of "FeDOM"? Seems to make better sense in the context of deconvoluting the particulate Fe phase into "FePOM" and "authFeOH" phases

Line 358: your study region might overlap the northern extent of the subtropical front, where higher PCa (from the Great Calcite Belt) is expected to contribute significantly to the particulate ("POM") pool. See papers by Barrett et al. (2018, Chemical Geology) and Balch et al. (various) about the Great Calcite Belt. Might be worth considering the role of Ca biominerals in future models of TM cycling, especially due to the effects of coccoliths as a particulate phase and on perturbations to total alkalinity and pH (which you note is a key parameter in the chemical speciation of Fe)

Figure 4 caption: "of" is repeated

Figure 5 symbols: difference between circle and triangle symbols might be difficult to see if plots are printed smaller, but I imagine that higher resolution versions will be available in digital format

Figure 6 caption: such a small detail, but perhaps swap the order of "Predicted" and "Observed" DFe in the caption to match the order of the distributions on the y-axis (where "Obs" appears before "Pred" sequentially)

Figure 7 y-axis scale: consider extending the y-axis scale to match the x-axis scale, so that the linear relationship between predicted and observed DFe is easier to see as nearly 1:1, aside from the acknowledged deviations in observed DFe at concs >2 nM. In other words, extend the y-axis to ~14 nM (but I leave the acceptance of this suggestion to the judgment of the authors)

Reviewer #2

(Remarks to the Author)

Gledhill and colleagues investigated the chemical speciation of dissolved and particulate Fe (at equilibrium) in the South Pacific Ocean using the ORCHESTRA program. Different set ups were tested that accounting for different Fe species. In the most comprehensive version this included (as I understand): free Fe', Fe bound to dissolved and particulate organic matter (accounting for heterogeneous affinities of different Fe binding sites), Fe bound to siderophores (one exemplary type), and authigenic Fe (i.e., precipitated Ferrihydrite). Inputs to this speciation model were taken from a very comprehensive dataset of parameters from a South Pacific Ocean transect and outputs (i.e., predicted dissolved and labile particulate Fe) were compared to observations to determine the realism of their model (i.e., Fe species and their characteristics, equilibrium assumption) for different parts of the transect and to investigate discrepancies from observations. I found this quite an interesting approach, however, I struggled to follow their description of the model (set-up), and, consequently, the interpretation of their results. Thus, I believe this manuscript would benefit from (major) revisions/clarifications to ensure accessibility to a broader audience.

Major comments:

The set up of the ORCHESTRA program is unclear to me. As I understand, it was set up in four different ways, 1) only accounting for FeDOM, 2) FeDOM + FeSid, 3) FeDOM + FeSid + authFeOH, 4) FeDOM + FeSid + authFeOH + FePOM. For exp. 1-3, these species sum up to match measured DFe concentrations, whereas for exp. 4 they sum up to TLFe (i.e., DFe + LPFe), with no prescribed partitioning between DFe and LPFe. If that's the case, does that mean that for experiments 1-3 all Fe species are (implicitly) assumed to be within the dissolved fraction or does the model also predict particle sized species in some way? If that is the case, was authFeOH considered to be part of the dissolved fraction in exp 3 but part of LPFe in exp. 4 (e.g., in Fig. 6)? Presumably it is present as both dFe and pFe? As for the functioning of the speciation (ORCHESTRA?) model – does it essentially account for concentrations and affinities (K') of different Fe'-competitors (DOM, POM, siderophores, precipitation) at the given environmental conditions and then calculate the corresponding Fe species concentrations (as a fraction of the total Fe, be it DFe or TLFe)? I think this needs to be made clearer. I was also confused about the role of the NICA-Donnan model – was that only used to estimate the (distribution of) DOM and POM K' 's? Is it part of the ORCHESTRA program? It would be very useful to the reader to include a more high-level description of the speciation model and experimental set up and, ideally, a schematic showing which input parameters and processes within the model (e.g., precipitation) were accounted for, and which outputs were then predicted (for each experiment or at least experiments 3 vs. 4). I had a look at the files in the provided data folders (especially the csv files), but the parameter names were not always clear to me and input and output variables should be better indicated.

For the presentation of the results, the use of different color scale ranges (and use of pmol vs. nmol) for different plots made comparison of predicted or measured Fe species very difficult. I would suggest using a uniform range of, e.g., 0-1 or 0-2 nM for the ODV plots and adding additional subplots showing the full ranges, if necessary. For some of the plots it could also be made clearer which parameters were observed vs. predicted, e.g., in Figures 3b-d (I assumed DFe was observed?). I also have some reservations regarding the use of absolute residuals for the comparison between predicted and observed dFe (or LPFe) species, as discussed in the specific below. Also, how come observed FeDOM (Fig. 2c) was never compared to predicted FeDOM (Fig. 4a)? Based on these plots, it seems like the model overestimates FeDOM quite substantially in some locations?

Because of my confusion regarding the methodological approach and the presentation of the results, it is not very clear to

me if the main finding that “accounting for the chemical heterogeneity of organic matter is essential for predicting widespread authFeOH formation” is supported. Would that not require an experimental set up where the heterogeneity of binding sites is not accounted, for comparison (i.e., to estimate the effect on authFeOH formation)? I am curious about how including DOM, POM, and siderophores affects such precipitation – does it increase the ‘apparent’ (?) solubility of Fe beyond what would be expected based on environmental conditions? It seemed to me that in the presented experiments (3 and 4) FeDOM and FePOM are mainly predominant in areas with low DFe (and low siderophores) where authFeOH is expected to be low/absent either way? There are a few other instances where I did not follow the authors’ interpretation of their findings, which I have indicated in the specific comments below.

Specific comments:

L. 47ff: “In biogeochemical models, ligand binding is typically too effective a competitor for Fe₂₇, so that Fe bound to ligands must be aggregated to POM via a “colloidal shunt”³”.

I am not sure what ‘too effective a competitor’ means here? Also, the “colloidal shunt” in ref. 3 describes the removal of colloidal-sized authigenic FeOH, not Fe bound to ligands (also applies to Fig. 1a, line 199)?

L. 50 ff.: “... the process can be reversible, with abiotic Fe remobilized from particles under certain conditions”
Can you specify these conditions? What about biological processes (e.g. remineralization or recycling)?

L. 52 ff.: “... in open ocean surface waters, Fe associated with particles is primarily biogenic, and POM has a higher mass fraction than lithogenic matter.”

I would specify open ocean waters without significant dust supply (or similar) here.

Figure 1b:

As mentioned above, I am unsure about the role of authFeOH – is it part of the labile particulate iron pool for all experiments or just the last one?

L. 72:

What are the stability constants used for the siderophores? Could you specify them here and, ideally, add them to in Figs. 3d, 5a-b for comparison?

L. 75: “whilst reactions that lead to authFeOH formation and reversible scavenging are represented by pathways #3 and #4.”
pathways #3 and #4, respectively? Or does this apply to both pathways? I am confused.

L. 102ff: “before our expedition”:

Please add the dates of the cruise here.

L. 108ff.: “whilst any longitudinal propagation”

Do you mean in east-west (zonal) direction (based on context)? I assumed (guessed) that “longitudinal” means in north-south direction. Or does the anti-cyclonic circulation prevent propagation in any direction? It would be helpful to indicate the predominant circulation patterns (as described here and in the following paragraph) in a figure (or added to Fig. 2c).

L. 132ff: “Here, we explore the chemical controls on observed DFe inventories, focusing on the role of organic matter and authFeOH production (Fig. 1b, pathway #1 to #3)”

Do you mean to say you first explore the controls on DFe (which is why only pathways 1-3 are listed)?

L. 135ff.: “we first estimated equilibrium partitioning of Fe between hydroxo complexes and DOM (pathway #1)”

To illustrate the effect on Fe’ of adding different pathways, I would suggest adding Fe’ distributions for each additional pathway (using the same color bar scale for comparison or by showing the difference to the “final” version shown in Fig. 3e). It would also help to add Figure S3a next to Figure 3a to see the effect of adding siderophores directly. Otherwise this first experiment (and even experiment 2) seem a bit futile.

L. 138ff.: “using a parameter set derived for marine DOM in the South Pacific³²”

I wonder if the parameters determined for the Peru margin zone are applicable to the entire South Pacific – did you confirm that?

L. 143ff.: “Dissolved Fe concentrations were the predominant influence on Fe’, but the heterogeneity of DOM binding sites led to a non-linear relationship between Fe’ and DFe concentrations, as the NICA model is underpinned by the Sips/Langmuir-Freundlich isotherm^{51,52}”

I would rephrase the first part of this sentence to something like “Fe’ increases with increasing Dissolved Fe but this relationship is non-linear (...),” since the statement “Dissolved Fe concentrations were the predominant influence on Fe’” seems a bit confusing? Also, I don’t know what the Sips/Langmuir-Freundlich isotherm is, so the second part is unclear to me, as well.

L. 158: “determined as their Fe complexes”

Does this mean that the siderophore concentrations represent siderophores bound to Fe? Shouldn’t this, ideally, be reproduced by the speciation model (i.e., no outliers from the 1:1 line in Fig. S3b)?

L. 170:

Can you clarify what apparent Fe solubility means in this context? Is this solely based on environmental conditions?

L. 172ff.: “supersaturation was also observed in proximity to the Peruvian shelf”

Are oxygen / Fe(II) concentrations and redox reactions accounted for in the speciation model? If not, this should be mentioned here (or at the end of this paragraph).

L. 179:

What does Fe concentration mean here (for this experiment)? Just DFe?

L.180ff.:

Does the range of affinities change between exp. 1-3 (i.e., if siderophores/Fe precipitation is accounted for)? Also how do they compare to the K' of siderophores (please add to figures, as discussed above)?

L. 193ff.: “Hence, binding site heterogeneity is essential for reconciling the dominance of organically bound Fe within the dissolved phase with the role of authFeOH production as a key mechanism for Fe removal from the ocean^{3,27}.”

Are you saying that the availability of a wide range of affinities of different binding sites explains why so much Fe (in the dissolved pool here) is associated with organic material? What about authFeOH associated with organic material? Also, when you say “the dominance of the organically bound Fe in the dissolved phase”, do you mean for this transect specifically or in general? The FeDOM values in Figure 2c do not seem particularly high compared to DFe in 2a (although it is difficult to compare them)?

L. 204:

What do you mean by preformed authFeOH?

L. 214ff: “We then assumed C comprised 50% of the POM mass to obtain the amount of POM in kg L⁻¹”

Is there a reference for this? This was quite confusing to me in general, since I was not aware that the speciation model (presumably) requires some input values in kg. This might be worth mentioning in a more general description of this model (in the supplement).

L. 215ff: “ Total labile Fe concentrations (TLFe) on incorporation of pathway #4 were set to the sum of DFe and LPFe and no prior assumptions were made with respect to partitioning between dissolved and particulate phases”

As discussed above, is TLFe the main input for the speciation model (i.e., do all Fe species in the output sum up to this)? If that is the case, it would be good to clarify this here.

L. 234. ff: “Occupied effective POM affinities (...) were approximately 1.7 log units higher than those of occupied DOM sites (...; Fig 5a). Higher affinities of occupied sites for the POM pool could reflect preferential partitioning of more hydrophobic (e.g. more unsaturated) DOM components into the POM phase and an increase in the molecular weight of POM in comparison to DOM^{39,73}.”

I did not understand this finding/did not follow this interpretation. Why are the predicted K' higher in the modelled POM than DOM? Does that relate to the model parameters or is this a consequence of the different POM, DOM vs. Fe concentrations or pH?

L. 250ff: “Nevertheless, calculations in the absence of siderophores confirmed that the trends in Fe:C ratios in DOM and POM were mainly driven by changes in the total Fe and C inventories through the water column.”

What does “calculations in the absence of siderophores” refer to? Also, can you clarify the second part of this statement (how changes in total Fe and C drive Fe:C ratios)? I am confused as to what else would drive the Fe:C ratios?

L. 253ff: “Although FePOM was the least abundant particulate phase (~2% of the LPFe inventory), it was a critical component when considering the LPFe probability density function(Fig. 6). “

What are the other phases of LPFe, is it just authFeOH? I'm also a bit confused about the density functions, do they show the distribution of observed/predicted concentrations across the transect, i.e., doesn't this figure just show that FePOM is important where LPFe is low? This could use some clarification. It might also be interesting to split up the predicted DFe into its components (FeDOM, FeSid, and Fe'?).

L. 255ff.: “Omission of POM binding from the calculations could not reproduce the observed density function and resulted in a shift in the density maxima for LPFe by close to an order of magnitude to 190 pmol L⁻¹ (Supplementary Fig. 5)”

What experiment is the dashed orange line in supp. Fig. 5 from? Could some of this predicted authFeOH be colloidal (i.e., not part of the LPFe pool)? It would be good to add the density function of LPFe observations to this figure for comparison.

L. 264: “At TLFe < ~1.5 nmol L⁻¹, however, no correlation was observed (Figure 4d inset)”

Do you mean Figure 7d? Does Figure 7c-d show the same relationships presented in Table 1? It might be helpful if the values in Table 1 were added to Figs. 7c-d instead (and if data 'in equilibrium' were indicated by using a different color in Fig. 7c-d). Also, does this mean at low concentrations (where FePOM is dominant) the predicted values are off?

L. 267ff: “across much of the SPO, predicted and observed concentrations matched closely (median residuals 0.031 and -0.002 nmol L⁻¹ for LPFe and DFe respectively)”

Shouldn't the median of absolute residuals (or similar) be used to estimate how close predictions match observations? It might also be interesting to see how large the residuals are at these locations relative to LPFe (i.e., what %), to ensure that you are not simply identifying locations with low overall LPFe (low overall Fe could also explain why there is little

authFeOH).

L. 269ff.: “We considered predicted concentrations within the interquartile range of LPFe (residual LPFe < -0.032 or > 0.157 nmol L⁻¹, n=201)

Should the “<” symbols be opposite here? Coloring this ‘equilibrium’ subset in Fig. 7d (as suggested above) would be useful for this, too.

L. 277ff.: “Our calculations suggest the scavenging residence time of particulate Fe is approximately 36 times longer than the particles (Supplementary Fig. 6).”

Is this referring to the correct supplementary figure?

L. 290: “Predicted DFe was lower than observed...”

I would add which species are summed up within predicted DFe (Fe', FeDOM, FeSid?) for easier interpretation

L. 296ff.: “Dissociation of Fe from DOM has a half-life of less than a day at 20°C...”

Are biological (recycling) processes considered in this? Do you have any data on microbial productivity for this transect? I found the interpretations in this paragraph quite hard to follow in general, I had assumed that the much higher observed DFe vs. predicted DFe (which only includes Fe', FeDOM, FeSid, as I understand) near the EPR and Peru margin was due to species not accounted for in the predicted DFe (or in the model), i.e., colloidal FeOH and/or Fe(II) species? Although I could be wrong, since I did not fully understand the speciation model set up (as discussed above). It does make sense to me that there's a kinetic control in some of these environments, but I don't know how relevant “authFeOH formation being the rate limiting step” is if there are various other Fe species that may potentially be involved (but not accounted for in the model or DFe prediction)?

L. 325ff.: “Accounting for the heterogeneity of organic matter binding site affinities was essential to reproduce Fe' concentrations consistent with authFeOH formation and prevents predictions of excess DFe that occurs if only one or two classes of binding sites are considered³.”

I think this is the main finding, but, as discussed above, I don't understand how/where the authors confirm this? Do the lower affinity binding sites (mentioned in the next sentence) set the Fe' levels that authigenic Fe can ‘access’? How is their effect (on “accessible” Fe') different from that of higher affinity sites / siderophores?

L. 337ff.: “when inorganic Fe concentrations become saturating”

What do you mean by inorganic Fe here?

Supplement:

L. 127ff: Stations 1-3, etc.

Please add these station numbers to a figure.

L. 140ff: “We therefore linearly interpolated between data points to a grid of 10 km by 1 m in R using the akima package” 10km distance and 1m depth? Seems a bit fine scale in the case of the FeSid dataset (might be worth mentioning as a limitation).

L. 142ff: “For pH, issues with the instrumentation meant that only a limited data set was available from the cruise itself, with better coverage towards the western part of the transect. We therefore used data from the P06 Clivar transect available in the GLODAPv2022 database¹⁹ as a base for interpolation of pH up to 146°W, ...”

Which GLODAP data was used for pH, was it calculated from alkalinity and DIC from the most the recent occupation (2017)? If that is the case, were changes (acidification) since 2017 accounted for (e.g., by comparing the 2010 and 2017 occupations or by comparing the GP21 data to the GLODAP data where available)?

L. 157ff: “We then assumed carbon comprised 50% of the POM mass to obtain the amount of POM in kg L⁻¹ (Supplementary Fig.2d).”

Is there a reference for this? Panel 2d does not exist?

Reviewer #3

(Remarks to the Author)

The authors present a framework for predicting dissolved and labile particulate Fe concentrations in the ocean interior (>250 m). The framework accounts for the formation of authigenic Fe (hydr)oxides, particulate organic Fe, and their interactions with dissolved organic matter and siderophores. A notable innovation is the use of a non-ideal competitive adsorption model (NICA) to estimate Fe–organic matter interactions, in contrast to the traditional competitive ligand exchange (CLE) approaches commonly used to quantify Fe–organic ligand binding. The proposed framework is compared against an existing model constructed around two broad ligand classes (strong and weak) coupled with a colloidal shunt mechanism. The authors argue that explicitly representing the chemical heterogeneity of organic matter, both dissolved and particulate, is essential for reproducing the observed vertical distributions of dissolved and labile particulate Fe in the South Pacific Ocean.

Overall, the manuscript presents a creative and theoretically grounded framework, supported by comparisons to measurements of dissolved and particulate Fe inventories. I commend the authors for advancing new conceptual tools for understanding the mechanisms that control Fe cycling in the ocean interior. The concepts covered in the manuscript are

compelling and worthy of publication. However, the technical density of the work makes it challenging to follow. It took several careful readings to fully distinguish the components of the framework, the measurements that were actually made, and how these map onto the modelled Fe pools. Some central elements of the framework, such as the NICA–Donnan implementation and the construction of the probability density functions, remain difficult to understand even with relevant background knowledge, and may prove inaccessible to the broad readership typical of Nature Communications.

Major Comments

1. Contextualizing the framework relative to existing models

The manuscript frames the new model primarily in contrast to the Tagliabue et al. (2023) framework (Fig. 1). My understanding is that the colloidal shunt was invoked largely to improve global biogeochemical model performance in regions such as the Atlantic, where high dust inputs and excess ligand concentrations lead to modeled dissolved Fe concentrations that exceed observations. This mechanism does not appear to be a dominant driver in the South Pacific. Therefore, the most meaningful distinction between the two frameworks in this study region lies in their respective treatments of organic matter–Fe interactions.

For this reason, it may be more appropriate to compare, or at least contextualize, the predictive skill of the new framework relative to contemporary global Fe biogeochemical models, in addition to their comparisons against observations. If the authors' contention is that this framework more accurately represents the controls of Fe in the ocean interior, a more explicit demonstration of its improvements over contemporary models would strengthen the argument.

2. Clarifying the role of DOM heterogeneity in driving authigenic Fe formation

One of the major conclusions is that the heterogeneity of DOM is essential for predicting widespread authigenic Fe formation. Setting aside the assumption that DOM functionality in the South Pacific is represented using DOM characterized on the Peruvian shelf, the mechanistic pathway by which these functional groups control authigenic Fe production is not clearly articulated.

It appears that this conclusion arises from the relatively low binding affinities assigned to the FeDOM pool via the NICA–Donnan model (compared to CLE-based estimates), and the assumption that particulate Fe above a threshold Fe:P ratio (converted to Fe:C and then to binding sites per kg C) represents authigenic Fe. However, the FePOM and FeAuth pools are operationally indistinguishable in the leaching method used for observations, and the division between them depends on an imposed threshold. This makes it difficult to evaluate how sensitive the conclusions are to separating these two pools.

Moreover, Fig. 7d shows strong predictability of LPFe when including samples with concentrations >1.5 nM (where authigenic Fe dominates), but much weaker predictive skill below this threshold. If DOM heterogeneity is the dominant control, one might expect predictive skill within the concentration regime where FeDOM is comparable to DFe (<1.5 nM). The fact that correlation is strongest when authigenic Fe dominates suggests that high LPFe inherently drives the agreement. If I am misinterpreting the authors' reasoning, a dedicated figure or explanatory paragraph clarifying how DOM heterogeneity leads mechanistically to the observed patterns would be helpful.

3. Acknowledging biological contributions to siderophores in deep-ocean Fe cycling

Although the manuscript focuses on abiotic transformations, biological processes also influence Fe cycling in the mesopelagic and deep ocean, which are an important control on multiple of the models' variables. Siderophores are microbially produced, and recent work shows active production in the mesopelagic (Li et al., 2024). Including siderophores in the model without acknowledging their biological origin leaves an important component of interpretation under-discussed.

Similarly, "reversible scavenging" likely includes processes such as biological uptake, remineralization, and re-packaging of Fe onto organic particles (Bressac et al., 2019; Bundy & Manck et al., 2025). Siderophore producers are also apart of the the FePOM pool and may influence residence times. For example, the observed longer residence time of FePOM relative to bulk particles could plausibly reflect biological Fe recycling. Even brief acknowledgment of these processes, especially where they may contribute to model uncertainty at low LPFe values, would strengthen the manuscript.

Furthermore, the model represents siderophores using ferrioxamine B, which is reasonable given limited binding constants for many siderophores to the cations in the model. However, the authors report detection of amphiphilic siderophore classes. Recent work indicates these are among the most ubiquitous in seawater (Li & Babcock-Adams et al., 2025). Amphibactins have substantially lower conditional stability constants than ferrioxamines (Bundy et al., 2018). I recommend that the authors consider a breakdown of ferrioxamine versus amphiphilic siderophore classes in their dataset and some commentary on whether inclusion of amphiphilic siderophores would alter the model results. Even a qualitative statement would be valuable.

Lines 165–166 suggest that Fe partitions into the FeDOM phase even when siderophore concentrations approach DFe concentrations. However, based on the ICP-MS method described, the authors' siderophore measurements directly detect ⁵⁶Fe-siderophore complexes, not apo-siderophores. High "siderophore" values therefore indicate Fe is partitioned into siderophore complexes when measured. However, in the model, Fe is predicted to partition into the FeDOM phase from DOC measurements, presenting a potential inconsistency. Does this mean the framework is misallocating Fe between organic pools?

4. Clarifying abbreviations, labels, and color scales between measured and modeled quantities

Throughout the manuscript, abbreviations and labels for Fe species need to be more clearly defined and consistently applied. For instance, if Fe' is being plotted (e.g., Fig. 3a), it should be labeled explicitly. Additionally, it is often unclear whether a plotted value is measured, modeled, or interpolated. Using siderophores as an examples:

-Fig. 2d appears to show measured FeSid

-Fig. 4b appears to show modeled FeSid

-Supplementary Fig. 2 shows interpolated siderophore concentrations

-Supplementary Fig. 3b plots "siderophore" vs "FeSid" without clear distinction

To avoid confusion, I recommend consistent, explicit notation (e.g., FeSid_meas, FeSid_model, FeSid_interp, FeDOM_NICA, etc.).

Predicted FeDOM values in Fig. 4 appear roughly twice (or more) as high as measured FeDOM values in Fig. 2, yet the color scales differ, making this difficult to see visually. Using consistent color ranges would help readers compare measured and modeled fields. Similarly, distinguishing measured vs predicted points through consistent symbol shapes or outlines would improve readability. More generally, for measurements spanning several orders of magnitude, log scaling may present the data more intuitively.

5. Clarifying model inputs and the construction of individual Fe pools and the probability density functions

For a broad readership, it is not sufficiently clear which measured parameters feed into each modelled Fe pool. A flowchart or summary table in the Supplementary Information detailing the inputs (e.g., DOC, DFe, TLFe, pH, T, siderophore measurements) for each of the model variables FeDOM, FeSid, FePOM, and FeAuth would improve accessibility.

The methodology underlying the probability density functions (PDFs) in Fig. 6 is not described in adequate detail. I can't find a clear explanation of the methods in the main text or Supplementary Information. As presented, the workflow appears circular: observations of dFe, LPFe, and FeDOM (modelled from DOC, pH, and temperature) are used to construct the PDFs, which then yield predictive value distributions that are compared back to the same observations. If this interpretation is incorrect, the manuscript should clearly describe the variables used to construct the PDFs, the statistical assumptions involved, and how independence is ensured between training and evaluation datasets.

Bressac, M., Guieu, C., Ellwood, M.J. et al. Resupply of mesopelagic dissolved iron controlled by particulate iron composition. *Nat. Geosci.* 12, 995–1000 (2019). <https://doi.org/10.1038/s41561-019-0476-6>

Li, J., Babcock-Adams, L., Boiteau, R.M. et al. Microbial iron limitation in the ocean's twilight zone. *Nature* 633, 823–827 (2024). <https://doi.org/10.1038/s41586-024-07905-z>

Bundy, R.M., Manck, L.E., Repeta, D.J. et al. Patterns of siderophore production and utilization at Station ALOHA from the surface to mesopelagic waters. *Limnol. Oceanogr.* (2024). <https://doi.org/10.1002/lno.12746>

Li, J., Babcock-Adams, L., McIlvin, M.R. et al. Distribution and cycling of siderophores in the Eastern North and Tropical Pacific Oceans. *Geophys. Res. Lett.* (2025). <https://doi.org/10.1029/2024GL113874>

Bundy, R.M., Boiteau, R.M., McLean, C. et al. Distinct siderophores contribute to iron cycling in the mesopelagic at Station ALOHA. *Front. Mar. Sci.* 5, 61 (2018). <https://doi.org/10.3389/fmars.2018.00061>

(Minor Comments)

Line 13 and Line 26 (and throughout): It seems that 'chemical/chemistry' and 'abiotic' are being used synonymously in this manuscript. Related to the large comment above, some chemistry is done by biology (siderophore production, Fe uptake, etc.). I recommend some more acknowledgement of biological controls in the language that is used.

Line 35: Would maybe expand on 'simple approximations' or at least clearly say what you are referring to.

Lines 40-54. Is biological uptake and settling included in this definition of 'scavenging'. Or 'binding to POM'?

Lines 117-126. I commend the authors on including the water mass analysis.

Lines 131-132. It would be nice to include correlation between AOU and DFe as a supplementary figure so that this relationship can be reviewed by readers

Lines 178-195. Can a more tangible statement be made to compare how the traditional CLE measurements of Fe-organic binding would differ from this NICA-Donnan model? There is a dataset from GP16 that could be used to provide a quantitative basis for comparison (Buck et al. 2018).

Buck, K.N., Sedwick, P.N., Sohst, B. and Carlson, C.A., 2018. Organic complexation of iron in the eastern tropical South Pacific: Results from US GEOTRACES Eastern Pacific Zonal Transect (GEOTRACES cruise GP16). *Marine Chemistry*,

201, pp.229-241.

Lines 234-241. More comparison to DOM knowledge may be warranted. One of the advantages of this approach, from my understanding, is modelling ligands from a more DOM-centric perspective. Given the same DOM binding parameters are applied to all depths and regions, it feels important to point out that the results are in line with what we know about molecular changes in DOM across ocean gradients. For example, do trends in modelled logK's align with how molecular indices or elemental compositions vary with depth or distance to coast? Osterholz et al., 2021 may be a relevant reference here, but there is a large body of literature on this topic.

Osterholz, H., Kilgour, D.P.A., Storey, D.S. et al. Accumulation of DOC in the South Pacific Subtropical Gyre from a molecular perspective. *Mar. Chem.* 231, 103955 (2021). <https://doi.org/10.1016/j.marchem.2021.103955>

Lines 263-264: The text references Figure 4d, but this should be Figure 7d.

Lines 325-326: "Accounting for the heterogeneity of organic matter binding site affinities was essential to reproduce Fe' concentrations consistent with authFeOH formation" This phrase reads as if Fe' and authFeOH were measured rather than inferred, the authors may want to restructure to clarify what they mean.

Supp. Lines 83-89. Are the DOM and siderophore recovery estimates used in the concentration calculations?

Reviewer #4

(Remarks to the Author)

Version 1:

Reviewer comments:

Reviewer #2

(Remarks to the Author)

The manuscript has been greatly improved and many of the issues I have raised in my previous review have been addressed, especially regarding the description of the methodologies used.

A couple of issues remain that I believe require addressing/additional clarification before publication, one regarding the representation of iron chemical speciation in models, another regarding the importance of binding site heterogeneity, plus a few minor issues/questions.

Figure 1a / L. 49ff.:

"In biogeochemical models, ligand binding is a strong competitor for Fe relative to the hydroxide ion, so that where Fe inputs are high, a portion of the DFe pool is aggregated to the particulate phase via a further empirically defined "colloidal shunt".

The second part of this sentence seems somewhat contradictory to the first part and might not be applicable to most biogeochemical models, as in most models ligands act to prevent DFe from precipitation. I believe the colloidal shunt pathway via authigenic colloidal Fe is exclusive to the model version described in Tagliabue et al 2023, as a way to explain substantial DFe removal despite undersaturated ligand pools.

I suggest to rephrase this sentence and update Figure 1 to either represent the approach of most (?) biogeochemical models:

- Ligands stabilise DFe (i.e., Fe' in equilibrium with FeL)
- Parts of Fe' is scavenged by particles (POM, dust, ...)
- If Fe' is in excess of inorganic Fe solubility (e.g., if ligand conc. < dFe conc.) it precipitates as authigenic Fe (i.e., is removed or moved to the particulate Fe pools)

or the approach of Tagliabue et al., 2023 (see Figure 2d of that paper), noting that this is unique to this model version:

- Fe' is in equilibrium with authigenic Fe (i.e., DFe in excess of Fe solubility is considered to be precipitated authigenic colloidal Fe)
- authigenic colloidal Fe partly aggregates into larger Fe particles and is removed ('colloidal shunt')
- Left-over soluble Fe is in equilibrium with (strong) ligands and stabilised, soluble Fe that is not bound to ligands can get scavenged

It might also make sense to include both approaches in Figure 1 to the reader's benefit.

L. 204ff. and Supplementary Figure 5

This paragraph is still a bit difficult to follow. As I understand (from the authors' response to my questions), the benefit of accounting for K' heterogeneity is that it is better at identifying the binding sites that outcompete Fe precipitation, and including heterogeneity therefore predicts more authigenic Fe formation than using average K' values (which overestimates the amount of binding sites that can compete). However, supplementary Figure 5 is a bit confusing – were the black open symbols calculated using eqn. 3-10 in the supplement or using ORCHESTRA? And was that also the case for the range of “19.7-23.1 after adjustment to relate to free Fe^{3+} ” on line 208 (or are those numbers taken from the original reference)? It would be good to refer to those supplementary equations where they were used. It would also be good to replace the black open symbols of supp. figure 5 with a different symbol so that it is easier to distinguish them from the colored symbols, and to more explicitly state in the figure caption that higher Fe' allows for more precipitation. The figure caption appears to be not very descriptive, overall, and could be shortened (e.g., by referring to eqn. 3-10, where appropriate).

I was originally hoping to see how much authigenic Fe is predicted to form (using ORCHESTRA) when heterogeneous binding sites are used vs. when only one or two classes of ligands with mean binding affinities were used. Presumably, the latter would be very sensitive to the mean affinity (i.e., a lot of authigenic Fe precipitation if the mean K' is below the minima shown in Fig. 4d but none if it is above)? Maybe this could be illustrated with some exemplary mean K' values? Also, are the values in Figure 4d the “mean” effective affinities (as described in the main text)? Please specify in caption.

Line 301: “median residuals 0.036 and -0.002 nmol L⁻¹ for DFe and LPFe”

To clarify - I meant that using absolute values (as in |residual|) when calculating the median would make more sense, as the median of positive and negative residuals could be 0 even if the individual residuals are large, as long as there is the same amount of positive and negative residuals.

Line 365ff.: “Hence it is particularly important to understand the role of weaker binding sites (...)”

I am not sure I follow this – isn't their role potentially negligible, if they are too weak to compete with precipitation? Is it not more important to be able to quantify the binding sites that are stronger than precipitation?

Line 375ff.:

“Considering POM binding sites as competitors for DOM creates an alternative or additional mechanism by which Fe can be “shunted” ”

Conceptually, wouldn't this be a form of scavenging? Based on Fig. 6, does this imply that scavenging by POM displays higher affinity to Fe' , on average, than ligands/DOM (except maybe siderophores)? This would be quite different from how scavenging is generally conceptualized/included in models (removing free Fe' not bound to ligands/not precipitated as authigenic Fe), which is interesting. Although based on the residence time discussion above (lines 306 ff.) wouldn't this be a continuous exchange ('reversible scavenging'), rather than a removal flux (i.e., a different effect than the 'colloidal shunt')? Do I interpret this correctly? The residence time discussion was still a bit confusing to me.

Line 419ff. (and 344ff.):

“We also identified a benthic input of LPFe east of the EPR, attributed to a supply of ferromanganese particles that are relatively inert (...)”

Could these particles also be responsible for the lower than predicted DFe (due to scavenging)?

ODV Figures:

Is it possible to adapt the colorbars to show only the first value of the final (yellow) color on top, e.g., 4 nmol/L in Figure 3 (i.e., so that it could be interpreted as any value ≥ 4 nmol/L is yellow)? In the current configuration it is often hard to tell which concentrations the different colors represent, as there are no labels at the lower levels where the colors are different.

Reviewer #3

(Remarks to the Author)

I appreciate that the authors have incorporated several of the suggestions raised during review. Overall, the manuscript presents an interesting and novel framework for predicting in situ Fe speciation and the formation of authigenic Fe phases in seawater. Developing alternative approaches for estimating free Fe and authigenic Fe formation is valuable and represents a useful contribution to the field.

While I have some reservations about the density and specialization of the topic with respect to the broad audience of Nature Communications, I believe the work is scientifically sound and worthy of publication. I therefore support publication of the manuscript after addressing one minor issue in the Supplementary Information noted below.

In Supplementary Figure 3, please clarify which data points represent direct measurements and which values were interpolated for use as model inputs. A much smaller subset of siderophore, DOC, and pH samples were directly measured compared to the number of values used in the interpolation. It would therefore be helpful to clearly indicate which samples were collected during this field campaign (as shown in Figure 2) and to describe in the caption what values or interpolation methods were used to generate the additional inputs for the model.

This could be addressed by using a distinct symbol to mark sampled points or by including an additional panel showing the spatial distribution of measured values relative to interpolated regions. Given that siderophores in the mesopelagic are known to exhibit highly patchy spatial distributions, distinguishing measured from interpolated values would improve the transparency and interpretation of this figure.

Reviewer #4

(Remarks to the Author)

Reviewer #1 (Remarks to the Author):

Outstanding manuscript and certainly "noteworthy"!

We very much appreciated the enthusiastic and supportive comments from reviewer #1. We are pleased that they fully appreciated both the significance and the potential impact of our work, given that we apply approaches and concepts that may be new to many ocean biogeochemists.

Direct answers to reviewer feedback prompts:

1. What are the noteworthy results? Gledhill et al. have synthesized a substantial data set of superior quality to produce a manuscript interpreting multiple interwoven processes into a cohesive description of Fe cycling in the South Pacific Ocean. Just the amount of analytical work required to produce the different data sets is herculean in scope and detail, but to then interpret all the concentrations, processes, and biogeochemical mechanisms in context...Nature Communications exists for publishing manuscripts like this!

2a. Will the work be of significance to the field and related fields? Yes! The interpretations and conclusions described in this manuscript help to conceptualize the major processes describing Fe biogeochemical cycling, with predictive power on Fe limitation and supply in diverse oceanographic regimes. In addition, the authors' approach to synthesizing multiple data sets provides a framework for other marine biogeochemists to interpret past and future data sets in the proper context of multiple related parameters.

2b. How does it compare to the established literature? The authors provide sufficient QA/QC values for all their parameters to allow direct comparison with past and future studies of one or more Fe data sets. Furthermore, the authors demonstrate the relevance of their research to the broader studies of biological and chemical oceanography, citing relevant publications that will allow future studies to discuss individual or multiple Fe data sets in the context of the "big picture."

3. Does the work support the conclusions and claims, or is additional evidence needed? The authors have provided EXCEPTIONAL evidence in support of their conclusions and claims. Seriously, each data set could be published individually and be noteworthy...seeing all these data sets and supporting data synthesized into a unified interpretation is beyond exemplary.

4. Are there any flaws in the data analysis, interpretation and conclusions? Do these prohibit publication or require revision? The authors sufficiently discuss any weaknesses in their synthesis, especially with respect to surface adsorption onto specific biominerals like silicate frustules or calcite tests. Instead of considering these acknowledgements as "flaws," they are properly noted as areas for future investigations, and do not in any way prohibit publication.

5. Is the methodology sound? Does the work meet the expected standards in your field? The methodologies and supporting QA/QC are exceptional, and only in a more focused publication on a single parameter would I expect to see additional analytical figures of merit.

6. Is there enough detail provided in the methods for the work to be reproduced? Absolutely! The authors give sufficient evidence in the supplemental materials for their own data quality as well as relevant references for their work to be duplicated.

I have only a few minor suggested revisions before publication, but I trust the editor(s) and authors to handle these minor updates without further input from me.

Lines 272-273: statement is a little awkward to understand. Do the authors mean that the solution and particles were governed by the concentration of Fe³⁺, POM, and ambient pH? Perhaps revise to read as:

"POM was thus the main carrier of Fe, where the exchange of Fe between solution and particles was governed by the concentration of Fe³⁺, POM, and the ambient pH"

We have changed the sentence as suggested (lines 304-305).

Line 336: change "deconvoluted" to "deconvolute"

Done (line 373)

Line 336: did you mean to write "FePOM" instead of "FeDOM"? Seems to make better sense in the context of deconvoluting the particulate Fe phase into "FePOM" and "authiFeOH" phases

Yes we did. Now corrected. (line 373)

Line 358: your study region might overlap the northern extent of the subtropical front, where higher PCa (from the Great Calcite Belt) is expected to contribute significantly to the particulate ("POM") pool. See papers by Barrett et al. (2018, Chemical Geology) and Balch et al. (various) about the Great Calcite Belt. Might be worth considering the role of Ca biominerals in future models of TM cycling, especially due to the effects of coccoliths as a particulate phase and on perturbations to total alkalinity and pH (which you note is a key parameter in the chemical speciation of Fe)

We found haptophytes made a minor contribution to the surface water phytoplankton community across our study region (Yuan et al., 2025) and examination of the Aqua-Modis Particulate inorganic carbon product for 2022 (<https://oceansdata.sci.gsfc.nasa.gov/l3/>, accessed 06.01.2026) indicates the

calcite belt lay further to the south (~40°S) at the time of our transect (March to April). Hence, we did not consider interactions with PIC in this study, but we agree this pathway should be considered in future studies.

Figure 4 caption: "of" is repeated

Corrected

Figure 5 symbols: difference between circle and triangle symbols might be difficult to see if plots are printed smaller, but I imagine that higher resolution versions will be available in digital format

We have left the symbols as triangles and circles since the two groups of data are mostly quite well separated within the plots.

Figure 6 caption: such a small detail, but perhaps swap the order of "Predicted" and "Observed" DFe in the caption to match the order of the distributions on the y-axis (where "Obs" appears before "Pred" sequentially)

This figure was removed following comments from reviewers #2 and #3 and replaced with a new figure in the SI (Supplementary Fig . 7) .

Figure 7 y-axis scale: consider extending the y-axis scale to match the x-axis scale, so that the linear relationship between predicted and observed DFe is easier to see as nearly 1:1, aside from the acknowledged deviations in observed DFe at concs >2 nM. In other words, extend the y-axis to ~14 nM (but I leave the acceptance of this suggestion to the judgment of the authors)

We think extending the y axis to ~14 nM is tricky as the data will only occupy the bottom 10% of the plot, which effectively obscures the data points that lie close to the 1:1 line. The 1:1 line should hopefully guide the viewer.

Reviewer #2 (Remarks to the Author):

Gledhill and colleagues investigated the chemical speciation of dissolved and particulate Fe (at equilibrium) in the South Pacific Ocean using the ORCHESTRA program. Different set ups were tested that accounting for different Fe species. In the most comprehensive version this included (as I understand): free Fe', Fe bound to dissolved and particulate organic matter (accounting for heterogeneous affinities of different Fe binding sites), Fe bound to siderophores (one exemplary type), and authigenic Fe (i.e., precipitated Ferrihydrite). Inputs to this speciation model were taken from a very comprehensive dataset of parameters from a South Pacific Ocean transect and outputs (i.e., predicted dissolved and labile particulate Fe) were compared to observations to determine the realism of their model (i.e., Fe species and their characteristics, equilibrium assumption) for different parts of the transect and to investigate discrepancies from observations. I found this quite an interesting approach, however, I struggled to follow their description of the model (set-up), and, consequently, the interpretation of their results. Thus, I believe this manuscript would benefit from (major) revisions/clarifications to ensure accessibility to a broader audience.

We thank the reviewer for their careful and insightful review of our manuscript and for highlighting the role of authFeOH colloids, which we had not incorporated within our discussion. We have clarified the model set up in a further figure (supplementary Fig 2) and improved the signposting and content for our more detailed description of how to use the model in our online data repository. We address further comments in detail below.

Major comments:

The set up of the ORCHESTRA program is unclear to me. As I understand, it was set up in four different ways, 1) only accounting for FeDOM, 2) FeDOM + FeSid, 3) FeDOM + FeSid + authFeOH, 4) FeDOM + FeSid + authFeOH + FePOM. For exp. 1-3, these species sum up to match measured DFe concentrations, whereas for exp. 4 they sum up to TLF_e (i.e., DFe + LPFe), with no prescribed partitioning between DFe and LPFe. If that's the case, does that mean that for experiments 1-3 all Fe species are (implicitly) assumed to be within the dissolved fraction or does the model also predict particle sized species in some way? If that is the case, was authFeOH considered to be part of the dissolved fraction in exp 3 but part of LPFe in exp. 4 (e.g., in Fig. 6)? Presumably it is present as both dFe and pFe?

Yes, this is the correct interpretation our approach. We added a simple schematic (supplementary Fig. 2) to clarify. We adopted a step-wise incorporation of the different competing components so that we could highlight the impact of each Fe' competitor under equilibrium conditions. Since the model is a chemical speciation model, it does not map directly onto physically defined phases, and some components are indeed likely present in both dissolved and particulate phases. The authFeOH component of the model is based on parameters derived for FeOH₃(s) formation which used a physical cutoff of 0.02 μm (Liu and Millero, 1999), hence authFeOH could be colloidal (greater than 0.02 but less than 0.2 μm). We have added the following text to the Methods in lines 189-191 to clarify this point *"In pathways #1-#3, we used DFe (<0.2 μm) as the input Fe concentration and predicted authFeOH could therefore be in the colloidal phase since the parameters on which authFeOH formation are based considered authFeOH to be >0.02 μm (Liu and Millero, 1999). Nevertheless, close to sources of Fe, it is also possible that the system is supersaturated with Fe' and has not yet reached equilibrium."*

In pathway 4 we use the LPFe as the input and we partition authFeOH into the particulate phase. The reviewer is right to highlight that colloids can also contribute to the systematic distribution of the residuals, and we have incorporated this information into the discussion in the following sentence *"Inputs from these sources are likely labile, dissolved and/or reduced forms of Fe (Fe(II))(Fitzsimmons et al., 2017; Gu et al., 2024; Tilliette et al., 2022), and we suggest these anomalies reflect kinetic control of LPFe-DFe partitioning via a colloidal authFeOH phase (<0.2 μm)"* (line 327).

As for the functioning of the speciation (ORCHESTRA?) model – does it essentially account for concentrations and affinities (K') of different Fe'-competitors (DOM, POM, siderophores, precipitation) at the given environmental conditions and then calculate the corresponding Fe species concentrations (as a fraction of the total Fe, be it DFe or TLF_e)? I think this needs to be made clearer. I was also confused about the role of the NICA-Donnan model – was that only used to estimate the (distribution of) DOM and POM K's? Is it part of the ORCHESTRA program? It would be very useful to the reader to include a more high-level description of the speciation model and experimental set up and, ideally, a schematic showing which input parameters and processes within the model (e.g., precipitation) were accounted for, and which outputs were then predicted (for each experiment or at least experiments 3 vs. 4). I had a look at the files in the provided data folders (especially the csv files), but the parameter names were not always clear to me and input and output variables should be better indicated.

Yes, the ORCHESTRA program (Meeussen, 2003) does account for concentrations and affinities – but the affinities are thermodynamic constants (i.e. $\log K$) that describe the intrinsic chemical properties of reactants. Hence our approach does not rely on empirical or conditional parameters (e.g. $\log K'$, ligand concentrations or an empirical definition of colloidal Fe). We believe this is a first attempt to mechanistically model these four pathways using an approach grounded in thermodynamics. Mechanistically grounded chemical processes have the potential to improve predictive power since changes in underlying environmental conditions such as pH and temperature can be fully accounted for, which is not the case for the empirical/conditional parameterisations used to date. We have rearranged the description of the speciation calculations to improve clarity. NICA-Donnan was used within ORCHESTRA to represent Fe binding to heterogeneous DOM / POM and was solved alongside the other equilibrium reactions (ion pairing, siderophore complexation and precipitation). We now renamed the section titled “**Chemical equilibrium calculations of multiple Fe species**” to “**Chemical equilibrium calculations for simultaneous prediction of multiple Fe species**” in the Methods (line 163). We started with the description of ORCHESTRA and added an overview of the approach with supplementary Fig 2. We added the following text to the methods (lines 165-169) to clarify “*For each sample, the model takes measured environmental conditions (major ions, pH, T and DOM/POM) together with a total Fe boundary condition ($D\text{Fe}_{\text{obs}}$ for pathways #1-#3, TLFe_{obs} for pathway #4) and returns the equilibrium distribution of Fe among the defined pools (Fe' , FeDOM , FeSid , authFeOH and FePOM), from which $D\text{Fe}_{\text{pred}}$ and $\text{LPFe}_{\text{pred}}$ are computed (Supplementary Fig 2).*”. The technical details of the ORCHESTRA program (Meeussen, 2003) are described extensively in the literature, hence we do not have to repeat them in this manuscript. We included a pdf and a Powerpoint slide show in the online repository (File name “Using Orchestra”) as a helper for those unfamiliar with the program. We moved the files to the top folder in the repository to enhance their visibility, added a higher-level schematic to this document and further explained input and output parameters in the pdf. We believe this is the most appropriate home for these technical details.

For the presentation of the results, the use of different color scale ranges (and use of pmol vs. nmol) for different plots made comparison of predicted or measured Fe species very difficult. I would suggest using a uniform range of, e.g., 0-1 or 0-2 nM for the ODV plots and adding additional subplots showing the full ranges, if necessary.

For some of the plots it could also be made clearer which parameters were observed vs. predicted, e.g., in Figures 3b-d (I assumed $D\text{Fe}$ was observed?). I also have some reservations regarding the use of absolute residuals for the comparison between predicted and observed $d\text{Fe}$ (or LPFe) species, as discussed in the specific below. Also, how come observed FeDOM (Fig. 2c) was never compared to predicted FeDOM (Fig. 4a)? Based on these plots, it seems like the model overestimates FeDOM quite substantially in some locations?

We adjusted predicted and observed $D\text{Fe}$, LPFe , FeDOM and Fe' plots (Figs 2, 3 and 4) to the same scale. We compared Fe:C ratios for observed and predicted FeDOM rather than FeDOM concentrations since observed FeDOM values are impacted by the low recoveries of DOM (ca. 11%, see methods) and we wished to avoid uncertainty around the assumption that FeDOM would be recovered with the same efficiency as all DOM. We clarified that FeDOM is not corrected for analytical recoveries in the caption of Fig. 2 and mention the low recovery of FeDOM in line 89 of the Methods and also added a statement here that neither FeDOM nor siderophores were corrected for recoveries (line 90-91, Methods & SI).

We used linear regressions to provide a measure of model performance (Table 1). Our use of absolute residuals in Fig. 8 was focused on identifying locations where predicted values do not match observed values and thereby pinpoint the mechanisms we have not captured in our model.

Because of my confusion regarding the methodological approach and the presentation of the results, it is not very clear to me if the main finding that “accounting for the chemical heterogeneity of organic matter is essential for predicting widespread authFeOH formation” is supported. Would that not require an experimental set up where the heterogeneity of binding sites is not accounted, for comparison (i.e., to estimate the effect on authFeOH formation)?

The reviewer raises an excellent point here. Fe' can be calculated from observations of discrete ligands and since there is an extensive dataset from the SPO (Buck et al., 2018), we have now used this data to illustrate the impact of not accounting for heterogeneity via addition of Supplementary Fig. 5. Application of only one or two mean conditional stability groups (Buck et al., 2018) to calculate Fe' results in Fe' concentrations that do not exceed its saturation state. This is despite the presence of elevated DFe due to inputs from reducing sediments and hydrothermal sources.

I am curious about how including DOM, POM, and siderophores affects such precipitation – does it increase the ‘apparent’ (?) solubility of Fe beyond what would be expected based on environmental conditions? It seemed to me that in the presented experiments (3 and 4) FeDOM and FePOM are mainly predominant in areas with low DFe (and low siderophores) where authFeOH is expected to be low/absent either way? There are a few other instances where I did not follow the authors’ interpretation of their findings, which I have indicated in the specific comments below.

The presence of DOM and siderophores is known to impact on Fe precipitation, since they compete with the hydroxide ion for Fe^{3+} . Hence they are acknowledged to increase the apparent Fe solubility e.g. (Liu and Millero, 2002; Zhu et al., 2021, 2023). Here we mechanistically explain why authFeOH is absent where DFe is low and suggest that POM binding sites can also participate in these competitive interactions, a factor which is not normally considered important in marine environments.

Specific comments:

L. 47ff: “In biogeochemical models, ligand binding is typically too effective a competitor for Fe, so that Fe bound to ligands must be aggregated to POM via a “colloidal shunt”³”.

I am not sure what ‘too effective a competitor’ means here? Also, the “colloidal shunt” in ref. 3 describes the removal of colloidal-sized authigenic FeOH, not Fe bound to ligands (also applies to Fig. 1a, line 199)?

We thank the reviewer for highlighting this point. We have rephrased this to “*ligand binding is a strong competitor for Fe (Tagliabue et al., 2019) relative to the hydroxide ion*” to clarify the meaning (lines 49-50). We have rephrased the text on the colloidal sized authFeOH (lines 50-51) so that it reads “*where Fe inputs are high, a portion of the DFe pool is aggregated to the particulate phase via an empirically defined “colloidal shunt”(Tagliabue et al., 2023)*”.

L. 50 ff.: “... the process can be reversible, with abiotic Fe remobilized from particles under certain conditions”

Can you specify these conditions? What about biological processes (e.g. remineralization or recycling)?

The abiotic remobilization of Fe from particles is typically associated with increased particle concentrations associated with benthic resuspension or hydrothermal plumes. We have rephrased the text to “*Whilst the term scavenging implies a removal or loss of Fe, the process can be reversible,*

with potential abiotic remobilization of Fe from particles when particle concentrations increase as a result of hydrothermal or sedimentary inputs (Achterberg et al., 2018; Boyd et al., 2017; Fitzsimmons et al., 2017) (lines 52-54). We discuss remineralization in lines 139-142, where we find a weak correlation between AOU and DFe within AAIW.

L. 52 ff.: "... in open ocean surface waters, Fe associated with particles is primarily biogenic, and POM has a higher mass fraction than lithogenic matter."

I would specify open ocean waters without significant dust supply (or similar) here.

We have clarified as suggested to "in open ocean surface waters that do not receive lithogenic particles from the atmosphere" (lines 56-57)

Figure 1b:

As mentioned above, I am unsure about the role of authFeOH – is it part of the labile particulate iron pool for all experiments or just the last one?

We have rearranged Fig. 1 to better reflect the potential size distribution of authFeOH and added the following text to the caption for 1 b) "Precipitated authFeOH could be present as colloids in the dissolved phase as well as in the particulate phase."

L. 72:

What are the stability constants used for the siderophores? Could you specify them here and, ideally, add them to in Figs. 3d, 5a-b for comparison?

We have clarified the stability constants used for the model siderophore ferrioxamine B "represented via stability constants **that describe binding of the model siderophore ferrioxamine B with protons, major ions and Fe**" (line 77). We do not think it is appropriate to add the constant for Fe binding to ferrioxamine B to the graphs as it is a thermodynamic constant ($\log K$ at 25°C = 36) (Schijf and Burns, 2016) and thus not comparable to the occupied effective affinities for DOM presented in the figures.

L. 75: "whilst reactions that lead to authFeOH formation and reversible scavenging are represented by pathways #3 and #4."

pathways #3 and #4, respectively? Or does this apply to both pathways? I am confused.

Respectively has been added to the sentence (line 82).

L. 102ff: "before our expedition":

Please add the dates of the cruise here.

The dates have been added (line 110).

L. 108ff.: "whilst any longitudinal propagation"

Do you mean in east-west (zonal) direction (based on context)? I assumed (guessed) that "longitudinal" means in north-south direction. Or does the anti-cyclonic circulation prevent propagation in any direction? It would be helpful to indicate the predominant circulation patterns (as described here and in the following paragraph) in a figure (or added to Fig. 2c).

We changed to zonal (line 116). Water masses are shown in Fig. 3c. It is challenging to show an anticyclonic circulation on a longitudinal (zonal) section plot since this is a horizontal anticlockwise

movement of water which restricts propagation in both zonal and meridional directions, but this can be readily observed in ref 44.

L. 132ff: “Here, we explore the chemical controls on observed DFe inventories, focusing on the role of organic matter and authFeOH production (Fig. 1b, pathway #1 to #3)”

Do you mean to say you first explore the controls on DFe (which is why only pathways 1-3 are listed)?
Yes, but we think the term “first” would be confusing here since we then step through pathways 1-3 in the subsequent paragraph.

L. 135ff.: “we first estimated equilibrium partitioning of Fe between hydroxo complexes and DOM (pathway #1)”

To illustrate the effect on Fe' of adding different pathways, I would suggest adding Fe' distributions for each additional pathway (using the same color bar scale for comparison or by showing the difference to the “final” version shown in Fig. 3e). It would also help to add Figure S3a next to Figure 3a to see the effect of adding siderophores directly. Otherwise this first experiment (and even experiment 2) seem a bit futile.

The reviewer makes an excellent point here. We have changed Fig. 3 to show the interpolated ODV plots of Fe' obtained after each pathway. We made a new Fig. 4 which shows the relationship between Fe' and DFe for pathways #1 to #3. The minimal difference between #1 and #2 or between #3 and #4 support our finding that addition of authFeOH has a critical impact on Fe' and thus Fe biogeochemistry, and we hope this finding is made clearer by consecutively stepping through each pathway.

L. 138ff.: “using a parameter set derived for marine DOM in the South Pacific³²”

I wonder if the parameters determined for the Peru margin zone are applicable to the entire South Pacific – did you confirm that?

Further work deriving NICA-Donnan parameters would certainly help to constrain how heterogeneity varies at the wider scale and we have added text to the limitations section to highlight this (lines 363-364). The parameter set was derived from titrations of multiple samples that exhibited a wide range of DOC concentrations (50-120 $\mu\text{mol L}^{-1}$) taken from both surface waters (<150 m) and deeper waters (up to 600 m). Zhu et al., (2021) did not detect any significant differences in Fe binding properties between surface and deep waters, which suggests even marked changes in DOC concentrations result in limited variability in parameters that describe DOM binding site heterogeneity. The molecular diversity of DOM across the SPO at depths >250 m is low (Osterholz et al., 2021), which further supports our use of one parameter set. We added the following text to the methods in lines 174-176: “Our use of one parameter set for the whole transect is supported by the low variability in DOM molecular composition reported for the SPO, especially for depths > 250 m (Osterholz et al., 2021).”

L. 143ff.: “Dissolved Fe concentrations were the predominant influence on Fe', but the heterogeneity of DOM binding sites led to a non-linear relationship between Fe' and DFe concentrations, as the NICA model is underpinned by the Sips/Langmuir-Freundlich isotherm^{51,52}”

I would rephrase the first part of this sentence to something like “Fe' increases with increasing Dissolved Fe but this relationship is non-linear (...)”, since the statement “Dissolved Fe concentrations were the predominant influence on Fe'” seems a bit confusing? Also, I don't know what the Sips/Langmuir-Freundlich isotherm is, so the second part is unclear to me, as well.

We have rephrased this sentence as suggested (lines 154-156) and moved the reference to the Sips/Langmuir isotherm to the methods (lines 208-216) together with the following explanatory text “The NICA component of the model is based on the Sips/Langmuir-Freundlich isotherm which is a modified version of the Langmuir isotherm (Koopal et al., 2020). The Langmuir isotherm is a fundamental model for surface adsorption processes (Langmuir, 1917) often applied in marine systems to calculate conditional stability constants and ligand concentrations (Gerringa et al., 1995). The Langmuir isotherm is suitable for describing interactions between one specific ligand and a metal. Application of the Langmuir isotherm to complex systems results in parameters that are difficult to interpret e.g. (Koopal et al., 2020; Perdue and Lytle, 1983). The Sips/Langmuir-Freundlich isotherm modifies Langmuir to account for interactions between a metal and multiple binding sites (ligands), where the properties of the individual binding sites cannot be determined because of the complexity of the system.”

L. 158: “determined as their Fe complexes”

Does this mean that the siderophore concentrations represent siderophores bound to Fe? Shouldn't this, ideally, be reproduced by the speciation model (i.e., no outliers from the 1:1 line in Fig. S3b)?

Thank you for raising this point. This was a technical detail and so we removed it to the SI (line 108). Since we detected siderophores mainly as their Fe complexes and approximated total siderophores from the concentrations of Fe complexes, the model output should reflect this. Less than 2% of the data points in Fig. S3 are outliers, which we consider well within expectations given analytical uncertainty and model simplifications.

L. 170:

Can you clarify what apparent Fe solubility means in this context? Is this solely based on environmental conditions?

Yes, it is based on environmental conditions. We removed the term “apparent” to avoid confusion.

L. 172ff.: “supersaturation was also observed in proximity to the Peruvian shelf”

Are oxygen / Fe(II) concentrations and redox reactions accounted for in the speciation model? If not, this should be mentioned here (or at the end of this paragraph).

We rephrased this paragraph to more clearly distinguish hydrothermal and Peru margin processes and inform the reader that Fe(II) is not included in the model. Lines 185-192 now read “The most supersaturated waters were associated with the highest observed DFe concentrations close to the Saguaro Vent on the EPR (Beaulieu and Szafranski, 2020; Hey et al., 2004) (Fig. 2), but supersaturation was also observed near the Monowai Volcano. Formation of authFeOH in proximity to hydrothermal systems is consistent with observations of authFeOH phases in particulate material isolated close to hydrothermal sources (Breier et al., 2012; Fitzsimmons et al., 2017; Heller et al., 2017; Marsay et al., 2018) and the role of authFeOH formation as an important mechanism for Fe removal from hydrothermal plumes (Tagliabue, 2014). Iron supersaturation was also observed in proximity to the Peruvian shelf, likely linked to inputs of reduced Fe (Fe(II)) from sediments (Gu et al., 2024; Scholz et al., 2014; Zhu et al., 2021). We note that redox processes are not accounted for in our model calculations.”

L. 179:

What does Fe concentration mean here (for this experiment)? Just DFe?

Yes, this has now been clarified via addition of Supplementary Fig. 2.

L.180ff.:

Does the range of affinities change between exp. 1-3 (i.e., if siderophores/Fe precipitation is

accounted for)? Also how do they compare to the K' of siderophores (please add to figures, as discussed above)?

The range in mean occupied affinities does indeed change between pathways #1 and #3. The upper limits of occupied affinities increases from $\log K_{\text{FeDOM}} = 22.6$ to $\log K_{\text{FeDOM}} = 23.7$ when the siderophore is added in pathway #2. This is because the high affinity siderophore binds a portion of the DFe, leaving only a fraction of the DFe pool to bind to DOM, which will then increase the mean affinity of the occupied FeDOM sites. In contrast, addition of pathway #3 results in an increase at the lower limit from $\log K_{\text{FeDOM}} = 15.6$ (pathways #1 and #2) to $\log K_{\text{FeDOM}} = 16.5$ as a result of authFeOH formation. We think a discussion of these points increases the technical complexity of the manuscript, but does not impact on the main findings, hence we prefer not to include this in the manuscript.

We think a comparison of conditional affinities determined for siderophores with occupied effective DOM affinities is also not beneficial since siderophores only occur in a limited region of the transect and are not identified as the main controls on Fe biogeochemistry. We added the following text to emphasize this point (lines 167-169) *"In addition to the ubiquitous $\text{FeDOM}_{\text{obs}}$ pool, we observed a patch of relatively high concentrations of siderophores (up to 425 pmol L^{-1}) at depths between 400 and 1500 m in the SPO gyre (Fig. 2d) centred around 117°W , suggesting microbial Fe limitation (Li et al., 2024) **although outside of this patch siderophores could not be detected**. Siderophores have a high affinity for Fe and will therefore compete with DOM binding sites **when present**."* Direct comparisons are also difficult since conditional stability constants for siderophores have only been determined for pH ca. 8 and temperatures of ca. $20\text{-}22^\circ\text{C}$ and not for ambient seawater conditions. We also have concerns about the accuracy of the K' for siderophores reported in the literature. Gerringa et al., (2021) found that none of the methods they applied in their paper to determine conditional stability constants of siderophores produced a constant that agreed with thermodynamic predictions for ferrioxamine B, likely because the method "detection windows" are too low to accurately represent the properties of this model siderophores. We feel an extensive (and likely highly technical) discussion of these factors would be more appropriately addressed in a different context, especially given the limited extent of siderophore occurrence across the SPO.

L. 193ff.: "Hence, binding site heterogeneity is essential for reconciling the dominance of organically bound Fe within the dissolved phase with the role of authFeOH production as a key mechanism for Fe removal from the ocean^{3,27}."

Are you saying that the availability of a wide range of affinities of different binding sites explains why so much Fe (in the dissolved pool here) is associated with organic material?

Our calculations indicate that where DFe concentrations are over 1.5 nmol L^{-1} , a fraction of the DFe pool is predicted to be present as authFeOH . We rephrased this sentence to *"Hence, binding site heterogeneity is essential for reconciling binding of Fe to DOM with the role of authFeOH production as a key mechanism for Fe removal from the ocean (Tagliabue et al., 2023, 2019)"* (lines 221-223). This result is not obtained when combining observations of conditional stability constants with ligand concentrations (as is current practice). We added results of calculations (supplementary Fig. 5) of Fe' from ligand concentrations and conditional stability constants to further expand on this point and clarify the impact of heterogeneity. In the caption of supplementary Fig. 5 we explain that *"Parameterisation of binding site heterogeneity allows for saturation of higher affinity binding sites and a decrease in the affinity of occupied binding sites (indicated by circle colour) as DFe increases. Higher affinity binding sites are thus "titrated out" until only weaker binding sites are in competition*

with the hydroxide ion and authFeOH can form. When Fe' is calculated from combining a conditional stability constant that is the mean of a group of ligands with the concentration of all ligands within each group, the many different binding sites within each group are assumed to have a uniform affinity for Fe - i.e. heterogeneity is not considered. For example, Fe' calculated from mean conditional stability constants (black diamonds) observed across the SPO at 16°S does not exceed its saturation state (Buck et al., 2018). Fe' is not predicted to be saturated even in close proximity to the Peruvian Shelf and at a prominent vent site at the EPR where DFe concentrations are higher than 10 nmol L⁻¹ (see inset) and authFeOH phases were observed in particles (Fitzsimmons et al., 2017). Combining a mean conditional stability constant with a total ligand concentration ignores the thermodynamic principle that chemical equilibria are determined by the affinity and concentrations of specific reactants. Parameterising the heterogeneity of DOM accounts for this principle and hence results in more accurate estimates of Fe' so that the formation of authFeOH can be predicted within a thermodynamically consistent framework.”

What about authFeOH associated with organic material?

It is quite possible that the authFeOH is associated with POM, but this interaction is not parameterized in the model, although it would certainly be interesting to add such an interaction in future studies if suitable parameters became available. We refer to the potential for such an association in the caption of Fig. 9.

Also, when you say “the dominance of the organically bound Fe in the dissolved phase”, do you mean for this transect specifically or in general? The FeDOM values in Figure 2c do not seem particularly high compared to DFe in 2a (although it is difficult to compare them)?

The determination of FeDOM in seawater is impacted by the poor recovery of DOM during preconcentration (recovery 11±4% at ambient pH, line 84-90, Methods), hence the FeDOM concentration in Fig 2c are not directly comparable to the predicted FeDOM concentration. We added the following explanatory text to the caption of Fig. 2: “Concentrations of FeDOM and FeSid are not corrected for analytical recovery (see Methods).” We do not correct for recoveries since we do not know if FeDOM is recovered with the same efficiency as DOC (now stated in Methods, line 90-91).

L. 204:

What do you mean by preformed authFeOH?

We removed this term.

L. 214ff: “We then assumed C comprised 50% of the POM mass to obtain the amount of POM in kg L⁻¹”

Is there a reference for this? This was quite confusing to me in general, since I was not aware that the speciation model (presumably) requires some input values in kg. This might be worth mentioning in a more general description of this model (in the supplement).

The NICA-Donnan model expresses binding site concentrations in terms of mass of organic matter (mol kg⁻¹ DOM), hence we must estimate this for the model. For DOM we had already stated the conversion factor in the Methods (line 176), we now added the relevant information for POM. We assumed 50% which is close to observed ratios of 53% for small particles (Lam et al., 2018, 2015). These two references have now been cited in the text in line 200.

L. 215ff: “ Total labile Fe concentrations (TLFe) on incorporation of pathway #4 were set to the sum of DFe and LPFe and no prior assumptions were made with respect to partitioning between dissolved and particulate phases”

As discussed above, is TLFe the main input for the speciation model (i.e., do all Fe species in the output sum up to this)? If that is the case, it would be good to clarify this here.

Yes this is correct, we rephrased the sentence (line 241-243) to clarify “For inclusion of pathway #4, total Fe concentrations were set to the sum of DFe_{obs} and $LPFe_{obs}$ (i.e. the total labile Fe concentration, $TLFe_{obs}$) and no prior assumptions were made with respect to partitioning between dissolved and particulate phases.” Mass balance is a convergence criterium for all speciation calculations, hence the output of all species must sum to the total input concentration.

L. 234.ff: “Occupied effective POM affinities (...) were approximately 1.7 log units higher than those of occupied DOM sites (...; Fig 5a). Higher affinities of occupied sites for the POM pool could reflect preferential partitioning of more hydrophobic (e.g. more unsaturated) DOM components into the POM phase and an increase in the molecular weight of POM in comparison to DOM_{39,73}.”

I did not understand this finding/did not follow this interpretation. Why are the predicted K' higher in the modelled POM than DOM? Does that relate to the model parameters or is this a consequence of the different POM, DOM vs. Fe concentrations or pH?

The POM parameterization (which drives this finding) is somewhat uncertain as we highlight in the limitations section, but even so it is clear that such a trend is necessary in order to explain the order of magnitude difference in Fe:C ratios between POM and DOM given the lower amounts of POM compared to DOM (with model Fe:C results consistent with observations as discussed in lines 273-277). We added the following phrase and reference (Lam and Xiang, 2025) since they provide an excellent explanation that expands on this point (lines 265-267): “Higher affinities of occupied sites for the POM pool are required to explain partitioning of Fe into the particulate pool **given the lower abundance of POM in comparison to DOM** (Lam and Xiang, 2025), but could also reflect preferential partitioning of more hydrophobic (e.g. more unsaturated) DOM components into the POM phase and an increase in the molecular weight of POM in comparison to DOM (Kharbush et al., 2020; Xu et al., 2020)”

L. 250ff: “Nevertheless, calculations in the absence of siderophores confirmed that the trends in Fe:C ratios in DOM and POM were mainly driven by changes in the total Fe and C inventories through the water column.”

What does “calculations in the absence of siderophores” refer to? Also, can you clarify the second part of this statement (how changes in total Fe and C drive Fe:C ratios)? I am confused as to what else would drive the Fe:C ratios?

We were interested in the impact of siderophores on Fe:C ratios in the OM pools. We rephrased the sentence to read “Preferential binding of Fe to the high affinity siderophore pool could be expected to reduce $Fe:C_{pred}$ within the OM pools and indeed the lowest calculated Fe:C ratios coincided with highest siderophore concentrations” to clarify (lines 279-281). We added supplementary Fig 2e to clarify the calculations undertaken in the absence of siderophores and explained the calculations in more detail in the caption.

L. 253ff: “Although FePOM was the least abundant particulate phase (~2% of the LPFe inventory), it was a critical component when considering the LPFe probability density function(Fig. 6). “

What are the other phases of LPFe, is it just authFeOH? I’m also a bit confused about the density

functions, do they show the distribution of observed/predicted concentrations across the transect, i.e., doesn't this figure just show that FePOM is important where LPFe is low? This could use some clarification. It might also be interesting to split up the predicted DFe into its components (FeDOM, FeSid, and Fe').

We have now simplified this part of the discussion and we hope the addition of Supplementary Fig 2 further clarifies the composition of LPFe. We removed the probability density functions from the manuscript and instead show the sensitivity of LPFe_{pred} to FePOM using a section plot (Supplementary Fig. 7).

The rewritten paragraph now reads *"We summed FeDOM_{pred}, FeSid_{pred} and Fe' to predict DFe concentrations (DFe_{pred}), and authFeOH and FePOM_{pred} to predict LPFe_{pred} concentrations. The major features in the DFe_{pred} and LPFe_{pred} distributions (Fig. 7a, b) reflected those of DFe_{obs} and LPFe_{obs} across the SPO (Fig. 2a, b). Predicted DFe correlated with DFe_{obs} (Table 1), though the relationship became weaker at higher concentrations (Fig. 7c). Predicted LPFe showed a strong correlation with LPFe_{obs} (Table 1), driven by authFeOH dominance at LPFe_{obs} > ~1.5 nmol L⁻¹ (Fig. 7d). We observed a weaker correlation between LPFe_{obs} and LPFe_{pred} when FePOM was the only LPFe_{pred} phase (n=113, p<0.001, r=0.2), likely a result of uncertainties in the parameterization of FePOM. Although FePOM_{pred} made up only ca. ~2% of the LPFe_{pred} inventory, it was a critical component when considering the LPFe_{pred} distribution (supplementary Fig. 7). Omission of POM binding from the calculations (Supplementary Fig. 2f) resulted in an absence of LPFe_{pred} for ca. 65% of the datapoints since concentrations of authFeOH were not predicted to be saturating. Hence a widespread LPFe_{pred} fraction could only be obtained when POM binding was included in the calculations."* (lines 284-295)

L. 255ff.: "Omission of POM binding from the calculations could not reproduce the observed density function and resulted in a shift in the density maxima for LPFe by close to an order of magnitude to 190 pmol L⁻¹ (Supplementary Fig. 5)"

What experiment is the dashed orange line in supp. Fig. 5 from? Could some of this predicted authFeOH be colloidal (i.e., not part of the LPFe pool)? It would be good to add the density function of LPFe observations to this figure for comparison.

We agree that some of the authFeOH could be colloidal and have more fully incorporated colloidal Fe into our discussion. We replaced Supplementary Fig. 5 with distributions of LPFe_{pred} (now Supplementary Fig. 7) to more directly illustrate the sensitivity of the results to the inclusion of the FePOM fraction.

L. 264: "At TLF_e < ~1.5 nmol L⁻¹, however, no correlation was observed (Figure 4d inset)" Do you mean Figure 7d?

We thank the reviewer for highlighting this and it is now corrected (line 289).

Does Figure 7c-d show the same relationships presented in Table 1? It might be helpful if the values in Table 1 were added to Figs. 7c-d instead (and if data 'in equilibrium' were indicated by using a different color in Fig. 7c-d).

We thank the reviewer for suggesting we highlight the equilibrium data with colour. The line in Fig. 7c and d shows the 1:1 relationship, not the relationship shown in Table 1. The 1:1 relationship allows a clear comparison of where observed and predicted concentrations do not agree, whilst the relationships in Table 1 are presented as figures of merit for model fit to data. We found addition of

the information in Table 1 to the figure resulted in crowded plots and hence we prefer to keep Table 1 separate.

Also, does this mean at low concentrations (where FePOM is dominant) the predicted values are off?

We reassessed our presentation of Fig. 7d and also our discussion of agreement between observed and predicted values at lower LPFe concentrations to provide a more robust comparison of model skill for FePOM versus authFeOH. Since we found that there is a weak correlation between predicted and observed values when only FePOM is present, we rephrased the paragraph to “We observed a weaker correlation between $LPFe_{obs}$ and $LPFe_{pred}$ when FePOM was the only $LPFe_{pred}$ phase ($n=113$, $p<0.001$, $r=0.2$), likely a result of uncertainties in the parameterization of FePOM. Although $FePOM_{pred}$ made up only ca. $\sim 2\%$ of the $LPFe_{pred}$ inventory, it was a critical component when considering the $LPFe_{pred}$ distribution (supplementary Fig. 7). Omission of POM binding from the calculations (Supplementary Fig. 2f) resulted in an absence of $LPFe_{pred}$ for ca. 65% of the datapoints since concentrations of authFeOH were not predicted to be saturating. Hence a widespread $LPFe_{pred}$ fraction could only be obtained when POM binding was included in the calculations.” (lines 289-295)

L. 267ff: “across much of the SPO, predicted and observed concentrations matched closely (median residuals 0.031 and -0.002 nmol L⁻¹ for LPFe and DFe respectively)”

Shouldn't the median of absolute residuals (or similar) be used to estimate how close predictions match observations?

We stated the median absolute residual, perhaps the reviewer did not spot this (line 301).

It might also be interesting to see how large the residuals are at these locations relative to LPFe (i.e., what %), to ensure that you are not simply identifying locations with low overall LPFe (low overall Fe could also explain why there is little authFeOH).

Presentation of relative residuals would be more prone to simply identifying locations with low overall LPFe, which is why we focused on absolute residuals. % residuals can be expected to increase at low LPFe because analytical uncertainty increases (affecting observations of LPFe) and because there are uncertainties in the FePOM binding parameters. Comparing Fig. 8b and Fig. 2b shows that absolute residuals are low where LPFe is low. Low overall LPFe does indeed explain why there is little authFeOH - our study provides the first mechanistic explanation grounded in thermodynamics for such a trend.

L. 269ff: “We considered predicted concentrations within the interquartile range of LPFe (residual $LPFe < -0.032$ or > 0.157 nmol L⁻¹, $n=201$)

Should the “<” symbols be opposite here? Coloring this ‘equilibrium’ subset in Fig. 7d (as suggested above) would be useful for this, too.

Yes (now corrected in line 302-303) – we thank the reviewer for spotting this and we coloured the equilibrium subset as suggested.

L. 277ff: “Our calculations suggest the scavenging residence time of particulate Fe is approximately 36 times longer than the particles (Supplementary Fig. 6).”

Is this referring to the correct supplementary figure?

This was an error – we did not mean to refer to a figure here. Furthermore, since supplementary Fig 6 was not referred to in the manuscript we removed it.

L. 290: "Predicted DFe was lower than observed..."

I would add which species are summed up within predicted DFe (Fe', FeDOM, FeSid?) for easier interpretation

This has been added (line 323).

L. 296ff: "Dissociation of Fe from DOM has a half-life of less than a day at 20°C..."

Are biological (recycling) processes considered in this? Do you have any data on microbial productivity for this transect?

This is an abiotic dissociation rate so does not consider microbial processes. Unfortunately, we also did not determine microbial productivity across the transect.

I found the interpretations in this paragraph quite hard to follow in general, I had assumed that the much higher observed DFe vs. predicted DFe (which only includes Fe', FeDOM, FeSid, as I understand) near the EPR and Peru margin was due to species not accounted for in the predicted DFe (or in the model), i.e., colloidal FeOH and/or Fe(II) species? Although I could be wrong, since I did not fully understand the speciation model set up (as discussed above). It does make sense to me that there's a kinetic control in some of these environments, but I don't know how relevant "authFeOH formation being the rate limiting step" is if there are various other Fe species that may potentially be involved (but not accounted for in the model or DFe prediction)?

We thank the reviewer for raising this point. We agree that colloidal Fe could also contribute to higher observed vs. predicted Fe and acknowledge we had not discussed this fully in the manuscript. We rephrased the text to clarify (lines 325-330). *"Inputs from these sources are likely labile, dissolved and/or reduced forms of Fe (Fe(II))(Fitzsimmons et al., 2017; Gu et al., 2024; Tilliette et al., 2022), and we suggest these anomalies reflect kinetic control of LPFe-DFe partitioning via a colloidal authFeOH phase. Since we allocate authFeOH to LPFe_{pred}, our calculations do not reflect the potential presence of a colloidal authFeOH phase (<0.2 μm) that would be an important intermediary during authFeOH aggregation (Tagliabue et al., 2023). Overall, the rate of formation of authFeOH will be set by the slowest reaction that occurs as an Fe source mixes with ambient seawater."* We think the information about the rate limiting step is important since it is always the rate of this step that determines the overall rate of a reaction and thus how long after perturbation (from e.g. Fe addition) a process will take to reach equilibrium.

L. 325ff.: "Accounting for the heterogeneity of organic matter binding site affinities was essential to reproduce Fe' concentrations consistent with authFeOH formation and prevents predictions of excess DFe that occurs if only one or two classes of binding sites are considered³."

I think this is the main finding, but, as discussed above, I don't understand how/where the authors confirm this? Do the lower affinity binding sites (mentioned in the next sentence) set the Fe' levels that authigenic Fe can 'access'? How is their effect (on "accessible" Fe') different from that of higher affinity sites / siderophores?

We fully acknowledge the reviewers point here. We have added supplementary Fig 5 to provide a comparison with Fe' calculated from ligands and conditional stability constants. This comparison demonstrates that representing DOM with a distribution of binding affinities, rather than a single mean conditional stability constant, results in Fe' values that are compatible with the onset of authFeOH formation. AuthFeOH formation (that is dependent on hydroxide ion concentration) determines the minimum affinity of the binding sites that can be occupied under the ambient conditions. Hence if this affinity is unknown then competition between authFeOH and binding sites

for Fe cannot be accurately determined. We rephrased this part of the paragraph (lines 360-365) to clarify “Accounting for the heterogeneity of organic matter binding site affinities was essential for a prediction of Fe’ concentrations consistent with authFeOH formation and accurate knowledge of the minimum affinity for binding sites that can be occupied under ambient conditions is necessary to constrain authFeOH formation. Hence it is particularly important to understand the role of weaker binding sites in Fe biogeochemistry and further investigations into how DOM heterogeneity varies with DOM abundance and lability (Lodeiro et al., 2023; Zhu et al., 2021) are required.”

Conditional stability constants represent a mean affinity (not the minimum) of all ligands (that were detected under the conditions of the experiment) and so cannot be used to accurately constrain competition between authFeOH formation and DOM.

L. 337ff.: “when inorganic Fe concentrations become saturating”

What do you mean by inorganic Fe here?

We rephrased this sentence to clarify. It now reads “The most critical pathways for constraining dissolved Fe phases were DOM (pathway #1) and authFeOH (pathway #3) since siderophores were constrained to the mesopelagic gyre and were not detectable in regions where $TLFe_{obs}$ concentrations were sufficiently high to result in authFeOH formation.”

Supplement:

L. 127ff: Stations 1-3, etc.

Please add these station numbers to a figure.

We thank the reviewer for spotting this. We do not show stations 1-3 on our figures as they were shelf stations, so the only relevant stations to mention here are station 22 at the EPR and 40 at the Monowai Volcano. We clarify which stations were omitted using their longitude rather than the station number (line 131).

L. 140ff: “We therefore linearly interpolated between data points to a grid of 10 km by 1 m in R using the akima package”

10km distance and 1m depth? Seems a bit fine scale in the case of the FeSid dataset (might be worth mentioning as a limitation).

There are three reasons why we think the resolution of siderophore interpolation, whilst high, is nevertheless suitable for our study. 1) we use the siderophore interpolation for conceptual representation of a particular binding site group in a model and not for the interpretation of siderophore distributions, 2) we need to use the same interpolation resolution for all data to match values across the dataset, 3) the interpolation is linear, hence utilizing a lower resolution would have a minor impact on the results.

L. 142ff: “For pH, issues with the instrumentation meant that only a limited data set was available from the cruise itself, with better coverage towards the western part of the transect. We therefore used data from the P06 Clivar transect available in the GLODAPv2022 database¹⁹ as a base for interpolation of pH up to 146°W, ...”

Which GLODAP data was used for pH, was it calculated from alkalinity and DIC from the most recent occupation (2017)? If that is the case, were changes (acidification) since 2017 accounted for (e.g., by comparing the 2010 and 2017 occupations or by comparing the GP21 data to the GLODAP data where available)?

We had inadvertently chosen to use the 2009 PO6 data to extend our pH values to the whole

transect, so we are grateful the reviewer raised this issue since we have been able to change the eastern part of the transect to the more recent 2017 data and more clearly specified the data we used in the Methods (line 147-148). We compared the 2009 and 2017 data but we found only minor differences (e.g. median (IQR) Fe' in east of 146°W was 217 (277) using 2009 pH data and 217 (286) using 2017 pH data) suggesting the slight drop in pH (<0.1 pH units) over this 8 year timescale does not impact our overall findings, although the impact over longer timescales could be more marked and worthy of investigation.

L. 157ff: "We then assumed carbon comprised 50% of the POM mass to obtain the amount of POM in kg L⁻¹ (Supplementary Fig.2d)."

Is there a reference for this? Panel 2d does not exist?

We added references and corrected the Figure number to Supplementary Fig 6b (line 161)

Reviewer #3 (Remarks to the Author):

The authors present a framework for predicting dissolved and labile particulate Fe concentrations in the ocean interior (>250 m). The framework accounts for the formation of authigenic Fe (hydr)oxides, particulate organic Fe, and their interactions with dissolved organic matter and siderophores. A notable innovation is the use of a non-ideal competitive adsorption model (NICA) to estimate Fe–organic matter interactions, in contrast to the traditional competitive ligand exchange (CLE) approaches commonly used to quantify Fe–organic ligand binding. The proposed framework is compared against an existing model constructed around two broad ligand classes (strong and weak) coupled with a colloidal shunt mechanism. The authors argue that explicitly representing the chemical heterogeneity of organic matter, both dissolved and particulate, is essential for reproducing the observed vertical distributions of dissolved and labile particulate Fe in the South Pacific Ocean.

Overall, the manuscript presents a creative and theoretically grounded framework, supported by comparisons to measurements of dissolved and particulate Fe inventories. I commend the authors for advancing new conceptual tools for understanding the mechanisms that control Fe cycling in the ocean interior. The concepts covered in the manuscript are compelling and worthy of publication. However, the technical density of the work makes it challenging to follow. It took several careful readings to fully distinguish the components of the framework, the measurements that were actually made, and how these map onto the modelled Fe pools. Some central elements of the framework, such as the NICA–Donnan implementation and the construction of the probability density functions, remain difficult to understand even with relevant background knowledge, and may prove inaccessible to the broad readership typical of Nature Communications.

We would like to thank the reviewer for their insightful comments and fully appreciate the extra mile the reviewer has travelled to comprehensively review our work. We agree the models and approaches we use may not be too familiar to many ocean biogeochemists, but we point out that the NICA–Donnan model is applied widely in the fields of aquatic sciences and geochemical modelling and was first developed in the 1990s (Benedetti et al., 1995). We were able to replace the probability density functions with a section plot that demonstrates the point we were trying to make (Fig. 6 and Supplementary Fig. 6 now replaced with Supplementary Fig. 7). Otherwise we aimed to supply sufficient technical information, using the SI for further detail, to allow results to be reproduced and for a thorough explanation of the approach. We thank the reviewer(s) for their contributions towards clarifying explanations and text within the manuscript. Overall, we think our ability to represent

chemical interactions within a thermodynamically consistent framework in a way which underpins recent emphasis on the contribution of authFeOH to Fe biogeochemistry (Tagliabue et al., 2023) is a critical finding that deserves a broad audience given the role of Fe as a key nutrient element in the ocean.

Major Comments

1. Contextualizing the framework relative to existing models

The manuscript frames the new model primarily in contrast to the Tagliabue et al. (2023) framework (Fig. 1). My understanding is that the colloidal shunt was invoked largely to improve global biogeochemical model performance in regions such as the Atlantic, where high dust inputs and excess ligand concentrations lead to modeled dissolved Fe concentrations that exceed observations. This mechanism does not appear to be a dominant driver in the South Pacific. Therefore, the most meaningful distinction between the two frameworks in this study region lies in their respective treatments of organic matter–Fe interactions.

We think Tagliabue et al., (2023) outlined an important conceptual step in our understanding of Fe biogeochemistry, as it was the first study to represent the impact of authFeOH formation in a global biogeochemical model. The focus of the Tagliabue et al., (2023) study was indeed on the North Atlantic. However, they applied an empirical formulation from Liu and Millero, (2002) to define colloidal authFeOH across the whole ocean. The SPO also receives elevated inputs of Fe from hydrothermal systems and shelf sources and hence there is quite a contrast between DFe at depths between their new PISCES-Quota-Fe and the traditional PISCES-Quota model (see extended data Fig. 4 in Tagliabue et al., (2023)) suggesting the colloidal shunt has an impact in our study region.

For this reason, it may be more appropriate to compare, or at least contextualize, the predictive skill of the new framework relative to contemporary global Fe biogeochemical models, in addition to their comparisons against observations. If the authors' contention is that this framework more accurately represents the controls of Fe in the ocean interior, a more explicit demonstration of its improvements over contemporary models would strengthen the argument.

We thank the reviewer for this suggestion. We have added a comparison with Fe' calculated from conditional stability constants and ligand concentrations determined across the SPO on GP16 (Buck et al., 2018) as supplementary Fig 5, which demonstrates the underlying issue that Fe' does not appear to be present at saturating concentrations if only one or two mean conditional stability constants are used in its calculation. The traditional way GBMs reduce ligand promoted accumulation of DFe is by setting a constant ligand concentration or to link ligands to DOC (Tagliabue et al., 2016), which allows ligands to become saturating at ca 0.6 nmol L^{-1} DFe and for excess DFe to be more efficiently removed from the system by scavenging. Nevertheless these traditional GBM approaches still resulted in elevated surface DFe (Tagliabue et al., 2016), hence the more recent addition of an empirical equation in the PISCES-Quota-Fe model to partition DFe into colloids that then aggregate to particulate Fe (Tagliabue et al., 2023). The underlying issue is one of accurately representing the competition between organic matter binding and authigenic iron formation. Unfortunately, the exact formulation used by Tagliabue et al., (2023) to distinguish colloidal and soluble Fe is not provided in the paper, which makes us reluctant to undertake a direct comparison between PISCES-Quota-Fe and our approach in this manuscript. However, for the purpose of this response we have made a comparison assuming soluble Fe (sFe) was calculated using Eq. 4 from Liu and Millero (2002) – such a

calculation is valid for seawater of ca. pH 8 with equivalent organic matter binding properties (which were left undefined in (Liu and Millero, 2002)) to those of the surface water gulf stream sample for which this equation was derived.

$$\log(sFe) = -10.53 + \frac{322.5}{T} - 2.5241 \cdot \sqrt{I} + 2.921 \cdot I \quad (1)$$

Where T is the temperature in K and I is the ionic strength. This equation can be used to calculate an empirical authFeOH fraction (authFeOH_{emp}) from the difference between DFe and sFe (Fig. R1). Overall, there is good agreement between authFeOH_{emp} and authFeOH predicted using the approach we adopt in this study. The empirical calculation of colloidal Fe captures an important process, which we mechanistically explain by accounting for DOM heterogeneity. Nevertheless, our estimates are off-set (by about 0.7 nmol L⁻¹) from those of the empirical approach (see Fig. R1). This off-set arises because most of the waters below 250 m in the SPO are below pH 8 and will also most likely contain less dissolved organic matter than the Gulf Stream surface waters used to derive Eq. 1. If we assume this is the equation used in Tagliabue et al., (2023) to calculate the colloidal fraction within their PISCES-Quota-Fe model, then we can follow the approach outlined in Fig. 3d of Tagliabue et al., (2023) and calculate Fe' from the reaction between soluble Fe and ligands. This results in estimates of Fe' that are several orders of magnitude lower than we calculate (Fig. R1b), which would reduce scavenging by particulate organic matter and minerals, since these processes depend on Fe' in the PISCES-Quota-Fe model (Aumont et al., 2015; Tagliabue et al., 2023).

Figure R1a) Comparison of authFeOH_{emp} calculated using an empirical approach valid for pH 8 and gulf stream surface water Fe binding properties (Liu and Millero, 2002) with authFeOH predicted in our study. b) Concentrations of Fe' (filled blue circles) calculated assuming only the soluble Fe fraction (i.e. DFe - empirical authFeOH) equilibrates with ligands following the approach outlined for PISCES-Quota-Fe in Tagliabue et al., (2023). Fe' predicted from pathways #1-#3 are shown as black diamonds for reference.

Irrespective of the details of the approach used in Tagliabue et al., (2023), it is acknowledged as empirical and authFeOH is stated to be out of equilibrium with the other Fe species in the system. Whilst we recognize the value of their findings and that empirical relationships are often essential in global biogeochemical models in order to reduce computational demand, adoption of an empirical approach for predicting authFeOH that does not account for the chemical process that underpins its formation risks bias in concentrations of the other interdependent Fe species that are components of

the model, and has the potential to overlook important changes in Fe solubility both in today's ocean and in the future ocean. We believe our approach represents a significant advance on current empirical approaches because it is thermodynamically consistent and represents the process of authFeOH precipitation from saturated soluble Fe hydroxides (i.e. Fe'). We have added clarifications to the manuscript that highlight this key aspect of our study.

In lines 71 we added "Here, we examine if authFeOH formation and the mechanisms underpinning Fe losses from the ocean could be constrained **within a thermodynamically consistent framework** that accounts for organic matter binding site heterogeneity (Lodeiro et al., 2023, 2019; Zhu et al., 2023) within the context of changes in ambient pH and temperature through the ocean water column"

In lines 91-93 we added "We found that these four **thermodynamically consistent** pathways can produce distributions of dissolved and labile particulate Fe that closely match observations"

In lines 409-410 we added the following sentence "Our results therefore support the role of authFeOH as a key controlling process regulating DFe concentrations in the ocean (Tagliabue et al., 2023, 2019) **and have constrained authFeOH production within a framework underpinned by the principles of thermodynamics**"

In lines 415-416 we add the following text "Accounting for binding site heterogeneity is critical for **predicting** authFeOH formation **within a thermodynamically consistent framework.**"

2. Clarifying the role of DOM heterogeneity in driving authigenic Fe formation

One of the major conclusions is that the heterogeneity of DOM is essential for predicting widespread authigenic Fe formation. Setting aside the assumption that DOM functionality in the South Pacific is represented using DOM characterized on the Peruvian shelf, the mechanistic pathway by which these functional groups control authigenic Fe production is not clearly articulated.

We thank the reviewer for highlighting this and agree that this point needed further clarification. We have added a new figure (Supplementary figure 5) that compares Fe' predicted when accounting for binding site heterogeneity with the NICA-Donnan model, with that predicted using observed parameters (the conditional stability constant and total ligand concentration) obtained across the SPO (Buck et al., 2018) that do not account for binding site heterogeneity. We also provide the following additional explanatory text to the Figure caption.

"Parameterisation of binding site heterogeneity allows for saturation of higher affinity binding sites and a decrease in the affinity of occupied binding sites (indicated by circle colour) as DFe increases. Higher affinity binding sites are thus "titrated out" until only weaker binding sites are in competition with the hydroxide ion and authFeOH can form. When Fe' is calculated from combining a conditional stability constant that is the mean of a group of ligands with the concentration of all ligands within each group, the many different binding sites within each group are assumed to have a uniform affinity for Fe - i.e. heterogeneity is not considered. The black diamonds show Fe' calculated from combining mean conditional stability constants with total ligand concentrations observed across the SPO at 16°S (Buck et al., 2018). Fe' does not exceed its saturation state even in close proximity to the Peruvian Shelf and at a prominent vent site at the EPR where DFe concentrations are higher than 10 nmol L⁻¹ (see inset) and authFeOH phases were observed in particles (Fitzsimmons et al., 2017). Combining a mean conditional stability constant with a total ligand concentration ignores the thermodynamic principle that chemical equilibria are determined by the affinity and concentrations

of specific reactants. Parameterising the heterogeneity of DOM accounts for this principle and hence results in more accurate estimates of Fe' so that the formation of authFeOH can be predicted within a thermodynamically consistent framework."

We also added the following explanatory text to the methods (lines 187-193): "Ferrihydrite precipitation was considered a proxy for authigenic Fe mineral (authFeOH) formation according to the equation(Liu and Millero, 1999)

hence in our model (pathway #3, Fig. 1b), **OM determines the concentration of Fe³⁺, which then precipitates when the solubility of Fe³⁺ (which is linearly related to concentrations of Fe) is exceeded.**"

It appears that this conclusion arises from the relatively low binding affinities assigned to the FeDOM pool via the NICA–Donnan model (compared to CLE-based estimates), and the assumption that particulate Fe above a threshold Fe ratio (converted to Fe:C and then to binding sites per kg C) represents authigenic Fe. However, the FePOM and FeAuth pools are operationally indistinguishable in the leaching method used for observations, and the division between them depends on an imposed threshold. This makes it difficult to evaluate how sensitive the conclusions are to separating these two pools.

Yes, the reviewer is correct, this conclusion does arise from the representation of a range of binding affinities, including those with lower affinities, afforded by application of the NICA-Donnan model. The reviewer is also right to point out that authFeOH and FePOM are not distinguished by the leach. In our model authFeOH and FePOM are distinguished by the chemical pathways by which their formation occurs and by the thermodynamic constants associated with those pathways. Hence, both fractions are determined by Fe' concentrations and calculated within a thermodynamically consistent framework. This means authFeOH and Fe:C ratios evolve as a result of the thermodynamic equilibria. We added the following to the caption of Fig. 1 "We represent four different **thermodynamically defined** pathways comprised of #1 reversible Fe binding to dissolved organic matter (DOM), #2 reversible Fe binding to microbial siderophores, #3 Fe precipitation as authFeOH and #4 reversible Fe binding to particulate organic matter". We have also reworked Fig. 1 and added supplementary Fig. 2 to further clarify the functioning of the model.

We evaluated the sensitivity of our results to the inclusion of the FePOM pathway to test the necessity of including both LPFe phases (lines 284-288), but we agree that this evaluation could have been more clearly expressed and following the reviewers' comments and suggestions have replaced the PDFs with a new supplementary Fig. 7, which contrasts the section plots for LPFe predicted with and without POM. We hope this demonstrates the role of FePOM in a more straightforward manner and shows that we need to include both fractions in order to fully account for partitioning between dissolved and labile particulate phases.

Moreover, Fig. 7d shows strong predictability of LPFe when including samples with concentrations >1.5 nM (where authigenic Fe dominates), but much weaker predictive skill below this threshold. If DOM heterogeneity is the dominant control, one might expect predictive skill within the concentration regime where FeDOM is comparable to DFe (<1.5 nM).

We thank the reviewer for highlighting this point and following the comment have re-examined our presentation of the results in this section. We found a clearer illustration of the relationship between

predicted and observed LPFe at low LPFe concentrations was obtained when we changed the scales of the inset in 7d to log-log to avoid overplotting. Following the reviewers comments we also revisited the way we had subset our data during the interpretation to more clearly focus on the relationship between predicted and observed LPFe when FePOM is the only LPFe phase calculated to be present. The text (lines 287-291) now reads *“Predicted LPFe showed a strong correlation with LPFe_{obs} (Table 1), driven by authFeOH dominance at LPFe_{obs} > ~1.5 nmol L⁻¹ (Fig. 7d). We observed a weaker correlation between LPFe_{obs} and LPFe_{pred} when FePOM was the only LPFe_{pred} phase (r=0.2, p<0.001, n=113), likely a result of uncertainties in the parameterization of FePOM.”*

The fact that correlation is strongest when authigenic Fe dominates suggests that high LPFe inherently drives the agreement. If I am misinterpreting the authors' reasoning, a dedicated figure or explanatory paragraph clarifying how DOM heterogeneity leads mechanistically to the observed patterns would be helpful.

We agree we had not linked DOM heterogeneity to the production of authFeOH very clearly. Following the suggestions of the reviewer, we have now added supplementary Fig 5 to clarify how DOM heterogeneity allows Fe' to become saturating and hence form authFeOH. We have further emphasized our approach seeks to understand the chemical processes that impact Fe biogeochemistry using a framework that is grounded in thermodynamic principles and thus avoids empirical approximations as outlined in our response to Major comment 1.

3. Acknowledging biological contributions to siderophores in deep-ocean Fe cycling

Although the manuscript focuses on abiotic transformations, biological processes also influence Fe cycling in the mesopelagic and deep ocean, which are an important control on multiple of the models' variables. Siderophores are microbially produced, and recent work shows active production in the mesopelagic (Li et al., 2024). Including siderophores in the model without acknowledging their biological origin leaves an important component of interpretation under-discussed.

The reviewer is correct to highlight that biological activity does control many of the model variables. We added the following sentence to highlight this in lines 142-143: *“Biological processes also influence pH and the distributions of organic matter, including siderophores, that determine the chemistry of Fe in seawater.”* We found siderophores were largely restricted to the mesopelagic, as did (Li et al., 2024) (discussed in lines 165-168). However, given siderophores were restricted to the mesopelagic gyre, where DFe and hence Fe' did not reach concentrations sufficiently high to result in authFeOH production, we prefer to keep a full discussion of the biological controls of siderophore production to a separate manuscript, where we can combine information on the different types of siderophores detected with further biological parameters that are relevant to the discussion.

Similarly, “reversible scavenging” likely includes processes such as biological uptake, remineralization, and re-packaging of Fe onto organic particles (Bressac et al., 2019; Bundy & Manck et al., 2025). Siderophore producers are also apart of the the FePOM pool and may influence residence times. For example, the observed longer residence time of FePOM relative to bulk particles could plausibly reflect biological Fe recycling. Even brief acknowledgment of these processes, especially where they may contribute to model uncertainty at low LPFe values, would strengthen the manuscript.

Our aim in this manuscript is to focus on the chemical aspects of Fe biogeochemistry, however we did not intend to give the impression that biological processes are not relevant to Fe biogeochemistry. We clarified our references to reversible scavenging as *“with potential abiotic*

remobilization of Fe from particles when particle concentrations increase as a result of hydrothermal or sedimentary inputs (Achterberg et al., 2018; Boyd et al., 2017; Fitzsimmons et al., 2017) so that it was more specific about the circumstances under which abiotic reversible scavenging could be critical. We also focused on depths >250 m in order to avoid depths where biological processes would be more intense, noting that biological regeneration within the SPO is shallow (<200 m) (Pavia et al., 2019). We added a clarification with respect to this in the limitations section “We restricted our analysis to depths below those expected to be strongly influenced by biological processes (Pavia et al., 2019) hence, in surface waters or the upper mesopelagic, where microbial remineralisation processes are intense, increased microbial activity could also influence Fe retention within the POM pool (Bressac et al., 2019; Bundy et al., 2025).” and cited the two suggested references.

Following the reviewers comments we also carefully considered the potential for a biological contribution to FePOM retention at depths >250 m. Fe:C ratios in prokaryotes are generally <100 $\mu\text{mol mol}^{-1}$ (Maldonado and Price, 1999; Pham et al., 2022), compared to our calculated mesopelagic Fe:C_{pred} ratios of ca. 400 $\mu\text{mol mol}^{-1}$. In deeper waters (>500 m) prokaryotes have been estimated to contribute < 10% of POC (Baltar et al., 2010). Prokaryotic Fe therefore likely comprises a minor component of the LPFe fraction at the depths our study considers. Thus, although bacteria in POM could take up and retain Fe, it seems unlikely that they are a major driver of observed Fe:C ratios within a POM pool dominated by detritus. Hence, we don't think we are able to link biological recycling to longer residence times for FePOM in this study.

We further examined the distribution DFe and LPFe residuals in the mesopelagic gyre, where interpolated siderophore concentrations were abundant (Fig. R2). Most samples within this region were identified as inliers and hence fell close to the 1:1 line for both DFe and LPFe and we observed no bias related to siderophore concentration. Hence, although we do not know if siderophores were abundant in the particulate phase and thus cannot directly assess the role of this potential pathway at the current time, our results suggest that the major trends in dissolved<->particulate partitioning within the siderophore patch are adequately explained by our model.

Fig. R2. Predicted and observed DFe (a) and LPFe (b) where interpolated siderophore concentrations were >0.1 nmol L⁻¹, and microbial siderophore production was thus likely to be most intense. The majority of data points within the siderophore patch were “inliers” (i.e. Equilibrium values shown as circles) with respect to LPFe residuals.

Furthermore, the model represents siderophores using ferrioxamine B, which is reasonable given limited binding constants for many siderophores to the cations in the model. However, the authors report detection of amphiphilic siderophore classes. Recent work indicates these are among the

most ubiquitous in seawater (Li & Babcock-Adams et al., 2025). Amphibactins have substantially lower conditional stability constants than ferrioxamines (Bundy et al., 2018). I recommend that the authors consider a breakdown of ferrioxamine versus amphiphilic siderophore classes in their dataset and some commentary on whether inclusion of amphiphilic siderophores would alter the model results. Even a qualitative statement would be valuable.

We added the following statement to lines 176-178 to acknowledge this point *“Furthermore, use of desferrioxamine B as the proxy siderophore potentially overestimates FeSid concentrations since desferrioxamine B has a high affinity for Fe relative to the amphiphilic marine siderophores that dominate mesopelagic environments (Bundy et al., 2018; Li et al., 2024).”*

Lines 165–166 suggest that Fe partitions into the FeDOM phase even when siderophore concentrations approach DFe concentrations. However, based on the ICP-MS method described, the authors’ siderophore measurements directly detect 56Fe-siderophore complexes, not apo-siderophores. High “siderophore” values therefore indicate Fe is partitioned into siderophore complexes when measured. However, in the model, Fe is predicted to partition into the FeDOM phase from DOC measurements, presenting a potential inconsistency. Does this mean the framework is misallocating Fe between organic pools?

We removed the technical detail relating to the siderophore method of determination to the SI (line 107), since this could have been confusing here. We found that observed FeDOM concentrations were $0.09 \pm 0.03 \text{ nmol L}^{-1}$ ($n=7$) where siderophore concentrations were $>0.1 \text{ nmol L}^{-1}$. Our model predicted FeDOM concentrations of 0.48 ± 0.27 ($n=78$) where $\text{FeSid}_{\text{pred}}$ was above 0.1 nmol L^{-1} . Keeping in mind that observed and predicted FeDOM are not directly comparable because of the likely poor analytical recoveries of FeDOM, we note that both observed and predicted results reflect the presence of excess DFe relative to siderophores. The model allocates Fe to the pools according to thermodynamic equilibria that are determined by the respective binding parameters. Our approach is therefore consistent for both types of Fe binding pool. Since the concentration of Sid is lower than that of Fe, and the high affinity siderophore ferrioxamine B is used as the siderophore analogue in the model, Sid is close to 100% bound to Fe. This is consistent with the mechanistic role of siderophores as a high affinity microbial Fe acquisition pathway. Nevertheless, since other marine siderophores are thought to bind Fe with lower affinities than ferrioxamine B, we added the following text to lines 176-178 *“Furthermore, use of desferrioxamine B as the proxy siderophore potentially overestimates FeSid concentrations since desferrioxamine B has a high affinity for Fe relative to the amphiphilic marine siderophores that dominate mesopelagic environments (Bundy et al., 2018; Li et al., 2024).”* We note that we observed very low abundance of apo-siderophores in our molecular ESI-MS data. Although this observation is not fully quantitative since apo and Fe bound siderophores have different sensitivities in ESI-MS, it does lend some support to our use of the FeSid concentrations as inputs for a (conceptual) total siderophore distribution in our model.

4. Clarifying abbreviations, labels, and color scales between measured and modeled quantities

Throughout the manuscript, abbreviations and labels for Fe species need to be more clearly defined and consistently applied. For instance, if Fe’ is being plotted (e.g., Fig. 3a), it should be labeled explicitly.

This has been checked and corrected where appropriate.

Additionally, it is often unclear whether a plotted value is measured, modeled, or interpolated. Using siderophores as an examples:

- Fig. 2d appears to show measured FeSid
- Fig. 4b appears to show modeled FeSid
- Supplementary Fig. 2 shows interpolated siderophore concentrations
- Supplementary Fig. 3b plots “siderophore” vs “FeSid” without clear distinction

To avoid confusion, I recommend consistent, explicit notation (e.g., FeSid_meas, FeSid_model, FeSid_interp, FeDOM_NICA, etc.).

We have added clarifying text to figures 2 and 4 in the manuscript and followed the reviewer’s suggestion to provide more explicit notation throughout the manuscript via addition of “pred” and “obs” subscripts where appropriate.

Predicted FeDOM values in Fig. 4 appear roughly twice (or more) as high as measured FeDOM values in Fig. 2, yet the color scales differ, making this difficult to see visually. Using consistent color ranges would help readers compare measured and modeled fields. Similarly, distinguishing measured vs predicted points through consistent symbol shapes or outlines would improve readability. More generally, for measurements spanning several orders of magnitude, log scaling may present the data more intuitively.

We have adjusted all scales so that figures are easier to compare to each other. Predicted FeDOM were indeed higher than observed FeDOM, since observations do not reflect the recovery of FeDOM, which is difficult to estimate. We clarified this by adding the following statement to the caption of Figure 2 and with further explanation within the methods.

Text added to caption of Figure 2: “Concentrations of FeDOM and FeSid are not corrected for analytical recovery (see Methods).”

Lines 90-91 in Methods: “Concentrations of FeDOM and siderophores were not corrected for recoveries.”

5. Clarifying model inputs and the construction of individual Fe pools and the probability density functions

For a broad readership, it is not sufficiently clear which measured parameters feed into each modelled Fe pool. A flowchart or summary table in the Supplementary Information detailing the inputs (e.g., DOC, DFe, TlFe, pH, T, siderophore measurements) for each of the model variables FeDOM, FeSid, FePOM, and FeAuth would improve accessibility.

We thank the reviewers for this suggestion. We added supplementary Figure 2 to detail inputs and show which processes are included in calculations. We also updated the information online to make the “using ORCHESTRA” instructions more clearly visible and clarified inputs and outputs in the document.

The methodology underlying the probability density functions (PDFs) in Fig. 6 is not described in adequate detail. I can't find a clear explanation of the methods in the main text or Supplementary Information. As presented, the workflow appears circular: observations of dFe, LPFe, and FeDOM (modelled from DOC, pH, and temperature) are used to construct the PDFs, which then yield predictive value distributions that are compared back to the same observations. If this interpretation is incorrect, the manuscript should clearly describe the variables used to construct the PDFs, the statistical assumptions involved, and how independence is ensured between training and evaluation datasets.

Following the reviewers suggestion we removed the PDFs from the manuscript since it is more straightforward to illustrate the sensitivity of $LP_{Fe_{pred}}$ to inclusion of FePOM with section plots (Supplementary Fig. 7).

Bressac, M., Guieu, C., Ellwood, M.J. et al. Resupply of mesopelagic dissolved iron controlled by particulate iron composition. *Nat. Geosci.* 12, 995–1000 (2019). <https://doi.org/10.1038/s41561-019-0476-6>

Li, J., Babcock-Adams, L., Boiteau, R.M. et al. Microbial iron limitation in the ocean's twilight zone. *Nature* 633, 823–827 (2024). <https://doi.org/10.1038/s41586-024-07905-z>

Bundy, R.M., Manck, L.E., Repeta, D.J. et al. Patterns of siderophore production and utilization at Station ALOHA from the surface to mesopelagic waters. *Limnol. Oceanogr.* (2024). <https://doi.org/10.1002/lno.12746>

Li, J., Babcock-Adams, L., McIlvin, M.R. et al. Distribution and cycling of siderophores in the Eastern North and Tropical Pacific Oceans. *Geophys. Res. Lett.* (2025). <https://doi.org/10.1029/2024GL113874>

Bundy, R.M., Boiteau, R.M., McLean, C. et al. Distinct siderophores contribute to iron cycling in the mesopelagic at Station ALOHA. *Front. Mar. Sci.* 5, 61 (2018). <https://doi.org/10.3389/fmars.2018.00061>

(Minor Comments)

Line 13 and Line 26 (and throughout): It seems that 'chemical/chemistry' and 'abiotic' are being used synonymously in this manuscript. Related to the large comment above, some chemistry is done by biology (siderophore production, Fe uptake, etc.). I recommend some more acknowledgement of biological controls in the language that is used.

We agree that biological processes result in DOM and POM and also influence pH, so the concentrations of all the substrates that are reacting with Fe are influenced by biological processes. Following the comment in major point 1 above we added the following statement: "Biological processes also influence pH and the distributions of organic matter, including siderophores, that determine the chemistry of Fe in seawater." to lines 142-143 and a clarification with respect to this in the limitations section (lines 377-380) "*We restricted our analysis to depths below those expected to be strongly influenced by biological processes (Pavia et al., 2019) hence, in surface waters or the upper mesopelagic, where microbial remineralisation processes are intense, increased microbial activity could also influence Fe retention within the POM pool (Bressac et al., 2019; Bundy et al., 2025)*".

Line 35: Would maybe expand on 'simple approximations' or at least clearly say what you are referring to.

We rephrased following sentence (line 36-38) to explain the approximations "*These approximations summarise Fe binding sites to one or two classes, typically termed "strong" and "weak", each with an empirically defined binding affinity, even though dissolved organic matter (DOM) is a heterogeneous*

pool of binding sites with a wide range of affinities (Boiteau and Repeta, 2022; Gledhill et al., 2022; Hassler et al., 2025; Li et al., 2024; Whitby et al., 2020)“

Lines 40-54. Is biological uptake and settling included in this definition of ‘scavenging’. Or ‘binding to POM’?

We do not consider that “scavenging” includes biological uptake – originally it was defined as removal of elements onto mineral hydroxides and oxides (Goldberg, 1954) and it is parameterised as an abiotic process in GBMs (e.g. Aumont et al., 2015). We added abiotic to this sentence to prevent any confusion “The main **abiotic** pathway for Fe removal from the ocean is its incorporation into particles that sink out of the water column, in a process broadly described as scavenging (Boyd et al., 2017; Tagliabue et al., 2023).” (lines 42-44). Likewise, “binding to POM” is a chemical process distinct from biological Fe uptake. We clarified the sentence to “Iron **chemically** bound to POM is likely exchangeable“ (line 54). We refer the reviewer back to their 3rd main comment above, where we discuss the contribution of prokaryotic C and Fe to the small non-sinking POM pool.

Lines 117-126. I commend the authors on including the water mass analysis.

We thank the reviewer for noting this.

Lines 131-132. It would be nice to include correlation between AOU and DFe as a supplementary figure so that this relationship can be reviewed by readers

We have added this figure to the SI (Supplementary Figure 1b)

Lines 178-195. Can a more tangible statement be made to compare how the traditional CLE measurements of Fe-organic binding would differ from this NICA-Donnan model? There is a dataset from GP16 that could be used to provide a quantitative basis for comparison (Buck et al. 2018).

Buck, K.N., Sedwick, P.N., Sohst, B. and Carlson, C.A., 2018. Organic complexation of iron in the eastern tropical South Pacific: Results from US GEOTRACES Eastern Pacific Zonal Transect (GEOTRACES cruise GP16). *Marine Chemistry*, 201, pp.229-241.

We thank the reviewer for this suggestion and have used this data set to demonstrate the role of binding site heterogeneity in supplementary Fig 5.

Lines 234-241. More comparison to DOM knowledge may be warranted. One of the advantages of this approach, from my understanding, is modelling ligands from a more DOM-centric perspective. Given the same DOM binding parameters are applied to all depths and regions, it feels important to point out that the results are in line with what we know about molecular changes in DOM across ocean gradients. For example, do trends in modelled logK’s align with how molecular indices or elemental compositions vary with depth or distance to coast? Osterholz et al., 2021 may be a relevant reference here, but there is a large body of literature on this topic.

Osterholz, H., Kilgour, D.P.A., Storey, D.S. et al. Accumulation of DOC in the South Pacific Subtropical Gyre from a molecular perspective. *Mar. Chem.* [231, 103955](https://doi.org/10.1016/j.marchem.2021.103955) (2021). <https://doi.org/10.1016/j.marchem.2021.103955>

We thank the reviewer for bringing this manuscript to our attention since the main finding that the molecular composition of DOM is similar across the SPO supports our use of one parameter set for Fe binding to DOM in the study area. We have cited the paper in the methods (line 174-176)

Lines 263-264: The text references Figure 4d, but this should be Figure 7d.
This has been corrected (line 289).

Lines 325-326: “Accounting for the heterogeneity of organic matter binding site affinities was essential to reproduce Fe’ concentrations consistent with authFeOH formation” This phrase reads as if Fe’ and authFeOH were measured rather than inferred, the authors may want to restructure to clarify what they mean.

We have rephrased to “Accounting for the heterogeneity of organic matter binding site affinities was essential for a prediction of Fe’ concentrations consistent with authFeOH formation” (lines 362-363)

Supp. Lines 83-89. Are the DOM and siderophore recovery estimates used in the concentration calculations?

We have clarified this point in the caption of Fig. 3 and a statement in the Methods (line 90-91). For siderophores, we thought the variability we determined for different siderophores was too high to apply a constant recovery, whilst for FeDOM, we only know the recovery of DOC, not the recovery of FeDOM, which makes it tricky to apply a correction factor.

Reviewer #4 (Remarks to the Author):

I co-reviewed this manuscript with one of the reviewers who provided the listed reports. This is part of the Nature Communications initiative to facilitate training in peer review and to provide appropriate recognition for Early Career Researchers who co-review manuscripts.
We thank the reviewer for their input.

Changes made to the manuscript to fit with Nature Communications editorial policy

In the abstract the final phrase edited to begin with “Our work shows”. Acronyms and abbreviations were removed

Figure captions have been edited to provide a brief title.

Response to reviewers:

R1 –

The manuscript has been greatly improved and many of the issues I have raised in my previous review have been addressed, especially regarding the description of the methodologies used.

A couple of issues remain that I believe require addressing/additional clarification before publication, one regarding the representation of iron chemical speciation in models, another regarding the importance of binding site heterogeneity, plus a few minor issues/questions.

We thank the reviewer for their positive comments and suggestions.

Figure 1a / L. 49ff.:

“In biogeochemical models, ligand binding is a strong competitor for Fe relative to the hydroxide ion, so that where Fe inputs are high, a portion of the DFe pool is aggregated to the particulate phase via a further empirically defined “colloidal shunt”.

The second part of this sentence seems somewhat contradictory to the first part and might not be applicable to most biogeochemical models, as in most models ligands act to prevent DFe from precipitation. I believe the colloidal shunt pathway via authigenic colloidal Fe is exclusive to the model version described in Tagliabue et al 2023, as a way to explain substantial DFe removal despite undersaturated ligand pools.

I suggest to rephrase this sentence and update Figure 1 to either represent the approach of most (?) biogeochemical models:

- Ligands stabilise DFe (i.e., Fe' in equilibrium with FeL)
- Parts of Fe' is scavenged by particles (POM, dust, ...)
- If Fe' is in excess of inorganic Fe solubility (e.g., if ligand conc. < dFe conc.) it precipitates as authigenic Fe (i.e., is removed or moved to the particulate Fe pools)

or the approach of Tagliabue et al., 2023 (see Figure 2d of that paper), noting that this is unique to this model version:

- Fe' is in equilibrium with authigenic Fe (i.e., DFe in excess of Fe solubility is considered to be precipitated authigenic colloidal Fe)
- authigenic colloidal Fe partly aggregates into larger Fe particles and is removed ('colloidal shunt')
- Left-over soluble Fe is in equilibrium with (strong) ligands and stabilised, soluble Fe that is not bound to ligands can get scavenged

It might also make sense to include both approaches in Figure 1 to the reader's benefit.

Following the suggestion, we rephrased the text to split the sentence and more clearly show that only the Tagliabue et al., (2023) model utilizes the colloidal shunt (lines49-52):

“In biogeochemical models, ligand binding is a strong competitor for Fe relative to the hydroxide ion²⁷ and thus limits authFeOH precipitation, which can lead to overestimation of DFe^{3,27}. Hence Tagliabue et al.,³ introduced an empirically defined colloidal shunt to aggregate a portion of the DFe pool to the particulate phase.”

We also rephrased the caption of Fig. 1a to “Chemical controls are currently described as binding between inorganic Fe (Fe') and ligands, scavenging onto particles, or precipitation as authigenic mineral

Fe (authFeOH). Tagliabue et al.,³ introduced a further colloidal authFeOH “shunt” that moves Fe from the dissolved to the particulate phase via aggregation.”

L. 204ff. and Supplementary Figure 5

This paragraph is still a bit difficult to follow. As I understand (from the authors’ response to my questions), the benefit of accounting for K’ heterogeneity is that it is better at identifying the binding sites that outcompete Fe precipitation, and including heterogeneity therefore predicts more authigenic Fe formation than using average K’ values (which overestimates the amount of binding sites that can compete). However, supplementary Figure 5 is a bit confusing – were the black open symbols calculated using eqn. 3-10 in the supplement or using ORCHESTRA? And was that also the case for the range of “19.7-23.1 after adjustment to relate to free Fe³⁺” on line 208 (or are those number taken from the original reference)? It would be good to refer to those supplementary equations where they were used. It would also be good to replace the black open symbols of supp. figure 5 with a different symbol so that it is easier to distinguish them from the colored symbols, and to more explicitly state in the figure caption that higher Fe’ allows for more precipitation. The figure caption appears to be not very descriptive, overall, and could be shortened (e.g., by referring to eqn. 3-10, where appropriate).

We have clarified and shortened the caption of Supplementary Figure 5. We changed the symbols to more clearly show the different data sets and we have also used a symbol shape to indicate data points where authFeOH precipitates to help emphasize that higher Fe’ leads to authFeOH precipitation. We have referred to equations 3-10 in the caption.

The caption now reads

“Supplementary Fig. 5. The impact of binding site heterogeneity on Fe’. Parameterisation of binding site heterogeneity results in a more pronounced increase in Fe’ as DFe increases (open symbols). The decrease in the mean affinity of occupied binding sites allows for more precise determination of the Fe’ concentration at which authFeOH precipitates (triangles indicate data points where authFeOH is predicted). When Fe’ is calculated from a mean conditional stability constant and a total ligand concentration, a uniform Fe affinity for binding sites is assumed, which overestimates the amount of binding sites that can compete with authFeOH formation. For example, Fe’ calculated from conditional stability constants and ligand concentrations (black crosses) observed across the SPO at 16°C¹⁶ using equations 2-9 (Supplementary Methods) does not increase sufficiently to exceed its saturation state, hence no authFeOH is predicted to precipitate. Inorganic Fe is not saturated even in close proximity to the Peruvian Shelf and at a prominent vent site at the EPR where DFe concentrations are higher than 10 nmol L⁻¹ (see inset) and authFeOH phases were observed in particles²¹.”

I was originally hoping to see how much authigenic Fe is predicted to form (using ORCHESTRA) when heterogeneous binding sites are used vs. when only one or two classes of ligands with mean binding affinities were used. Presumably, the latter would be very sensitive to the mean affinity (i.e., a lot of authigenic Fe precipitation if the mean K’ is below the minima shown in Fig. 4d but none if it is above)? Maybe this could be illustrated with some exemplary mean K’ values? Also, are the values in Figure 4d the “mean” effective affinities (as described in the main text)? Please specify in caption.

Whilst this approach is appealing, it is indeed very sensitive to parameter selection – and would require selection of both suitable binding affinities and suitable ligand concentrations. We think it would be difficult to meaningfully constrain parameters for such an exercise, given that data on which any estimates might be based are only applicable to ca. 20°C and pH 8. Hence we chose to use data from GP16 and

equations 2-9 (Supplementary Methods), since this approach is not reliant on concentrations of other pH and temperature dependent Fe species.

Line 301: “median residuals 0.036 and -0.002 nmol L⁻¹ for DFe and LPFe”

To clarify - I meant that using absolute values (as in |residual|) when calculating the median would make more sense, as the median of positive and negative residuals could be 0 even if the individual residuals are large, as long as there is the same amount of positive and negative residuals.

We do not expect the median of positive and negative residuals to be zero because observed DFe/LPFe are not inputs to the model. Hence there are systematic offsets between the modelled DFe/LPFe values and the observed DFe/LPFe values that reflect the way in which DFe and LPFe are defined in the model and uncertainties in the model parameterization and it these offsets we wished to explore through this calculation.

Line 365ff.: “Hence it is particularly important to understand the role of weaker binding sites (...)”

I am not sure I follow this – isn't their role potentially negligible, if they are too weak to compete with precipitation? Is it not more important to be able to quantify the binding sites that are stronger than precipitation?

We replaced “weaker binding sites” with “role of binding site heterogeneity” (line 312).

Line 375ff.:

“Considering POM binding sites as competitors for DOM creates an alternative or additional mechanism by which Fe can be “shunted” ”

Conceptually, wouldn't this be a form of scavenging? Based on Fig. 6, does this imply that scavenging by POM displays higher affinity to Fe', on average, than ligands/DOM (except maybe siderophores)? This would be quite different from how scavenging is generally conceptualized/included in models (removing free Fe' not bound to ligands/not precipitated as authigenic Fe), which is interesting. Although based on the residence time discussion above (lines 306 ff.) wouldn't this be a continuous exchange ('reversible scavenging'), rather than a removal flux (i.e., a different effect than the 'colloidal shunt')? Do I interpret this correctly? The residence time discussion was still a bit confusing to me.

The reviewer makes a good point with respect to this concluding sentence and we rephrased the sentence to reflect the primary role of POM as a reversible scavenger. The sentence now reads (line 321-322): “The POM phase represents reversible scavenging and its response to changes in pH, temperature, TLF_e concentration and POM abundance”.

Line 419ff. (and 344ff.):

“We also identified a benthic input of LPFe east of the EPR, attributed to a supply of ferromanganese particles that are relatively inert (...)”

Could these particles also be responsible for the lower than predicted DFe (due to scavenging)?

We did not observed a correlation between observed DFe and LPMn within the datapoints with negative LPFe anomalies ($r = 0.09$, $p = 0.37$, $n = 11$) hence we do not have any evidence to support a significant scavenging effect of ferromanganese particles.

ODV Figures:

Is it possible to adapt the colorbars to show only the first value of the final (yellow) color on top, e.g., 4 nmol/L in Figure 3 (i.e., so that it could be interpreted as any value ≥ 4 nmol/L is yellow)? In the current configuration it is often hard to tell which concentrations the different colors represent, as there are no labels at the lower levels where the colors are different.

We added three contours to the plots in Fig. 3 to enhance the readability within the low concentration range.

R3 –

I appreciate that the authors have incorporated several of the suggestions raised during review. Overall, the manuscript presents an interesting and novel framework for predicting in situ Fe speciation and the formation of authigenic Fe phases in seawater. Developing alternative approaches for estimating free Fe and authigenic Fe formation is valuable and represents a useful contribution to the field.

While I have some reservations about the density and specialization of the topic with respect to the broad audience of Nature Communications, I believe the work is scientifically sound and worthy of publication. I therefore support publication of the manuscript after addressing one minor issue in the Supplementary Information noted below.

We thank the reviewer for their suggestions.

In Supplementary Figure 3, please clarify which data points represent direct measurements and which values were interpolated for use as model inputs. A much smaller subset of siderophore, DOC, and pH samples were directly measured compared to the number of values used in the interpolation. It would therefore be helpful to clearly indicate which samples were collected during this field campaign (as shown in Figure 2) and to describe in the caption what values or interpolation methods were used to generate the additional inputs for the model.

This could be addressed by using a distinct symbol to mark sampled points or by including an additional panel showing the spatial distribution of measured values relative to interpolated regions. Given that siderophores in the mesopelagic are known to exhibit highly patchy spatial distributions, distinguishing measured from interpolated values would improve the transparency and interpretation of this figure.

We included an additional panel in Supplementary Figure 3 that allows for a comparison between the observations and the interpolation.

R4 –

I co-reviewed this manuscript with one of the reviewers who provided the listed reports. This is part of the Nature Communications initiative to facilitate training in peer review and to provide appropriate recognition for Early Career Researchers who co-review manuscripts

We thank the reviewer for their input.